# Learning to Reason and Act in Cascading Processes Driven by Semantic Events

## Abstract

Training agents to control a dynamic environment is a fundamental task in AI. In many environments, the dynamics can be summarized by a small set of events that capture the semantic behavior of the system. Typically, these events form chains or cascades. We often wish to change the system behavior using a single intervention that propagates through the cascade. For instance, one may trigger a biochemical cascade to switch the state of a cell or reroute a truck in logistic chains to meet an unexpected, urgent delivery.

We introduce a new supervised learning setup called *Cascade*. An agent observes a system with known dynamics evolving from some initial state. It is given a structured semantic instruction and needs to make an intervention that triggers a cascade of events, such that the system reaches an alternative (counterfactual) behavior. We provide a test-bed for this problem, consisting of physical objects.

We combine semantic tree search with an event-driven forward model and devise an algorithm that learns to efficiently search in exponentially large semantic trees of continuous spaces. We demonstrate that our approach learns to effectively follow instructions to intervene in new complex scenes. When provided with an observed cascade of events, it can also reason about alternative outcomes.

## 1 Introduction

Teaching agents to understand and control their dynamic environments is a fundamental problem in AI. It becomes extremely challenging when events trigger other events. We denote such processes as *cascading processes*. As an example, consider a set of chemical reactions in a cellular pathway. The synthesis of a new molecule is a discrete event that later enables other chemical reactions. Cascading processes are also prevalent in man-made systems: In assembly lines, when one task is completed, e.g., construction of gears, it may trigger another task, e.g. building the transmission system.

Cascading processes are abundant in many environments, from natural processes like chemical reactions, through managing crisis situations for natural disasters (Zuccaro et al., 2018; Nakano et al., 2022) to logistic chains or water treatment plants (Cong et al., 2010). A major goal with cascading processes is to intervene and steer them towards a desired goal. For example, in biochemical cascades, one hopes to control chemical cascades in a cell by providing chemical signals; in logistics, a cargo dispatch plan may be completely modified by assigning a cargo plane to a different location.

This paper addresses the problem of reasoning about a cascading process and controlling its qualitative behavior. We describe a new counterfactual reasoning setup called "*Cascade*", which is trained via supervised learning. At *inference* time, an agent observes a dynamical system, evolving through a cascading process that was triggered from some initial state. We refer to it as the "unsatisfied" or "observed" cascade. The goal of the agent is to steer the system toward a different, counterfactual, configuration. That target configuration is given as a set of qualitative constraints about the end results and the intermediate properties of the cascade. We call these constraints the "instruction". To satisfy that instruction, the agent intervenes with the system at a specific point in time by changing the state of one element which we call the "pivot".

To solve the *Cascade* learning problem, we train an agent to select an intervention given a state of a system and an instruction. Importantly, we operate in a counterfactual mode (See Pearl, 2000). During training, the agent only sees scenarios that are "satisfied", in the sense that the system dynamics

Figure 1: **An experimental test bed for the *Cascade* setup**. **Input 1** (the unsatisfied cascade): A set of balls is observed *moving* in a confined space, colliding with each other, with walls, and with static pins (grey & black). Collisions yield a cascade of events (arrows). **Input 2**: A complex instruction describes a desired "counterfactual" cascade of events and its constraints. **Output** (the satisfied cascade): The agent intervenes and sets the (continuous, 2D) initial velocity of the purple ball (the "pivot") to achieve the goal, *satisfying the constraints*. Only keyframes are shown. See full videos here: `https://youtu.be/u1Io-ZWC1Sw` (Anonymous)

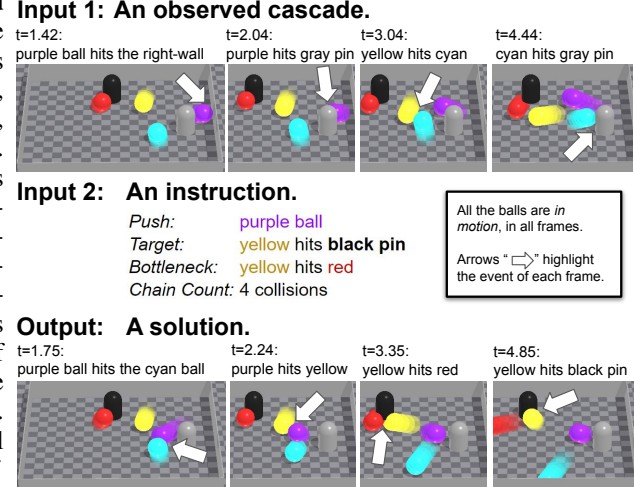

obey the constraints given in the instruction. The reason is that in the real world it is not possible to rewind time and simultaneously obtain both a satisfied and an unsatisfied sequence of events.

**Steering a cascade process is hard.** To see why, consider a natural but naive approach to the Cascade problem: train an end-to-end regression model that takes the system and instruction as input and predicts the necessary intervention. It is challenging because in many cases, a slight change in one part of the system can make a qualitative effect on the outcome. This may lead to an exponential number of potential cascades. This "butterfly effect" (Lorenz, 1993) is typical in cascading systems, like a billiard ball missing another ball by a thread or a truck reaching a warehouse right after another truck has already left. Back to the regression approach, we empirically find that it fails, presumably because the set of possible chains of events is exponentially large, and the model fails to learn how to find an appropriate chain that satisfies the instruction. We discuss other challenges in Section 4.

**Technical insights.** In designing our approach, we follow two key ideas. First, instead of modeling the continuous dynamics of the system, we reduce the search space by focusing on a small number of discrete, semantic events. To do this, we design a representation called an "*Event Tree*" (Figure 2). In a billiard game, these events would be collisions of balls. In logistic chains, these events would be deliveries of items to their target location or assembly of parts. To reduce the search space, we build a tree of possible future events, where the root holds the initial world-state. Each child node corresponds to a possible future subsequent event from its parent. Thus, a path in the tree from a root to a descendant captures a realizable sequence of events.

Our second idea is to learn how to efficiently search over the event tree. This is critical because the tree grows exponentially with its depth. We learn a function that assigns scores to tree nodes conditioned on the instruction and use these scores to prioritize the search. We also derived a Bayesian correction term to guide the search with the observed cascade: we first find the path in the event tree that corresponds to the observed cascade, and then correct the scores of nodes along that path.

**Modelling system dynamics with forward models.** A forward model describes the evolution of the dynamic systems in small time steps. There is extensive literature on learning forward models from observations in physical systems (Fragkiadaki et al., 2016; Battaglia et al., 2016; Lerer et al., 2016; Watters et al., 2017; Janner et al., 2019). Recent work also studied learning forward models for cascades (Qi et al., 2021; Girdhar et al., 2021). However, once the forward model has been learned, the desired initial condition of the system is found by an exhaustive search. Here, we show that exhaustive search fails for complex cascades and with semantic constraints (Section 5). Therefore, our paper focuses on *learning to search* not on *learning the forward model*. We assume that we are given a special kind of a "forward" model operating at the level of semantic events. Namely, given a state of the cascading system, our forward model allows to query for the next event ("which objects collide next?"), and predict the outcome of that event (velocities of objects after collision).

**Test bed.** We designed a well-controlled environment that shares key ingredients with real-world cascading processes. In our test-bed several spheres move freely on a table, colliding with each other

and with static pins within a confined space (Figure 1). The chain of collisions forms a complex cascading process. Naturally, a simulated test-bed cannot cover the full complexity of real-world scenarios and additional research may be required. We discuss these topics in Section 7 .

**Contributions.** This paper proposes a novel approach for learning to *efficiently search* for a complex cascade in a dynamical system. Our contributions are: (1) A new learning setup, *Cascade*, where an agent observes a dynamical system and then changes its initial conditions to meet a given semantic goal. (2) Learning a principled probabilistic scoring function over an – *Event Tree* – data structure, for searching efficiently over the space of interventions. (3) A Bayesian formulation leveraging the observed cascade to guide the search in the event tree toward a counterfactual outcome.

## 2 RELATED WORK

**Learning and reasoning in physical systems.** Several papers studied cascading events in the context of physical systems. There, the main focus was to use object interactions to learn a forward model from observations. PHYRE, Virtual Tools, and CREATE (Bakhtin et al., 2019; Allen et al., 2020; Jain et al., 2020) are benchmarks for physical reasoning for computer vision. They differ from our learning setup in three key aspects. First, the current paper focuses on the *search* problem, looking to satisfy a set of semantic constraints on the event sequence. Second, in the prior benchmarks, all tasks have to satisfy the same final goal, rather than being conditioned on a semantic goal. Last, their setup is a sequential decision reinforcement learning setup, allowing exploration, collecting rewards from the environment, and multiple retries, which are not allowed in our setup. In addition, no event-driven forward model (EDFM) is currently available for these benchmarks, and training an EDFM requires additional annotations and is beyond the scope of this work. There are several approaches to learn such models from temporal data, like dynamic Bayes nets (Bhattacharjya et al., 2020; Ghahramani, 1998; Gunawardana & Meek, 2016), which can also handle latent variables.

Allen et al. (2020) takes a Bayesian approach for updating the distribution of initial conditions given a reward. We consider the underlying chain of events and update the value of the node scores in the event tree according to the observed cascade. CLEVRER, CoPhy, CRAFT, CATER, and IntPhys (Yi et al., 2020; Baradel et al., 2020; Ates et al., 2021; Girdhar & Ramanan, 2020; Riochet et al., 2018) are benchmarks for reasoning over observed temporal and causal structures in video. They differ from our setup in a few key aspects: (1) They focus on video-tracking and question answering rather than acting . (2) In CLEVRER and CoPhy, the observed cascade is available during training, which may not be a reasonable assumption for real-world problems (see Section 3). (3) CoPhy estimates the value of a *static* observed  property, like gravity, while we focus on the cascade evolution.

Roussel et al. (2019) studied the chain reaction problem, with a different focus than ours. Their cascading configuration is fully given and they study how to tune that configuration using a simulator. Our work focuses on *finding* a cascading configuration given a partial description of it.

**Graph Neural Networks** have been used in physical environments (Kim & Shimanuki, 2019; Shen et al., 2020; Bapst et al., 2019), representing the underlying state as a graph, without considering temporal consequences. Temporal consequences are vital for our decision system. We propose how to transform a *temporal sequence* of events to a DAG.

**Reinforcement learning:** The "Cascade" learning setup is fundamentally different from a reinforcement learning framework (Sutton & Barto, 2005). The problem we try to solve is not a standard planning problem (Hafner et al., 2019), where a series of actions are taken sequentially. Here, an action is taken once and sets the cascade of events ("Fire and forget").

**Planning in robotics:** Pertsch et al. (2020); Jayaraman et al. (2019) learned from video data to predict key-frames, conditioned on a start frame and an end frame (goal). These works rely on a visual end goal. It is unclear how to use them with a semantic goal that includes constraints. They also rely on taking multiple actions, which is not applicable in our "Fire and forget" setup.

**Causal inference:** Counterfactual reasoning was studied in causal inference (Pearl, 2000). Most relevant is (Buesing et al., 2019) that used  counterfactually augmented data for training a RL policy.

## 3 THE "*Cascade*" LEARNING SETUP

*Cascade* is a supervised learning problem. At the *inference* phase, The agent is provided with a dynamical system and two inputs: (1) A sequence of events called the "observed cascade" together

with the respective initial condition of the system. (2) An instruction that describes desired semantic properties ("constraints") of the solution. The observed cascasde does not satisfy the instruction. The agent is asked to intervene by controlling the state of one "pivot element" in the system at a specific point in time. The goal is to find an intervention that makes the roll out of the dynamical system satisfy the instruction.

At the *training* phase, we are *only given* "successful" labeled examples. The "features" (x) of each label consists of (1) an instruction; and (2) the initial state of the system except the controllable pivot. The "label" (y) of each sample is the initial state of the pivot, which yields the desired behavior of the system. During training we do not provide examples of failing sequences together with a successful sequence. The reason is that in reality, one cannot "roll-back" time and obtain both a failed sequence and a successful sequence.

More formally, our training set $\mathcal{D}$ consists of $N$ labelled samples $\mathcal{D} = \{\text{features} = (x_n, g_n), \text{label} = (y_n^*, Q(y_n^*), n = 1 \ldots N\}$, where:

- $x_n$ is the initial state of a dynamical system, excluding those of its pivot element.
- $g_n \in \mathbb{R}^G$ is a structured representation of an instruction.
- $y_n^* \in \mathcal{Y} \subset \mathbb{R}^d$ is the pivot's initial state of the solution.
- $Q(y_n^*) = \{s_k\}_{k=1}^K$ is a sequence of events that occur when the system is played out with the pivot initial value $y_n$.

*At test time*, a novel sample is drawn, describing an unseen dynamical system and instruction $(x, g)$; and an observed cascade roll-out $(y^{obs}, Q(y^{obs}))$ which fails to fulfill the instruction. Our goal is to provide an alternative (counterfactual) initial state for the pivot element $\hat{y}$, such that the instruction is fulfilled when the system is rolled-out.

**Our test bed:** We introduce a new simulated test bed that abstract away from specific applications. An agent observes a physical world with several moving and static objects going through a cascade of events (Figure 1 top), and it is given a *complex* instruction "*Push: purple ball . . .*". It then manipulates the direction and speed of the purple ball (Figure 1 bottom) manifesting a new cascading process that satisfy the complex set of constraints given by the instruction. In the *Cascade* setup, the agent is trained on a set of scenes and their goals, and is tested on new scenes and their goals.

## 4 METHODS

Solving the Cascade setup poses three major challenges. First, our model needs to match semantic events, but simulations of dynamical systems typically follow fixed and small timesteps, which are indifferent to events. Second, the set of desired constraints and dynamical systems is compositional and large. The agent should learn to generalize to different systems and configurations that were not observed during training. Third, we wish to benefit from examples of failed cascades that are available at inference time (the counterfactual setup).

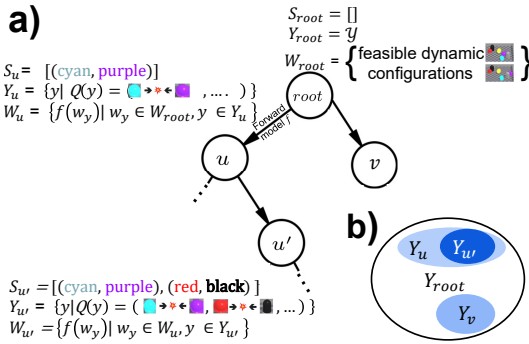

Figure 2: **(a)** The *Event Tree* data structure, illustrated according to our test-bed. $S$ is the collision sequence of a node; $Y$ is the intervention subset of a node; $W$ is the node's world-state. See Section 4.1. **(b)** Tessellation of the intervention space.

We develop an approach that addresses these three challenges. To address the first, we develop a representation that focuses on key "semantic" events of the dynamics (e.g., collisions). We build a tree of possible outcomes such that a path in the tree captures a realizable cascade of events. To address the second challenge, we learn a scoring function that assigns values to tree nodes conditioned on the instruction. This allows us to generalize to unseen setups, and at inference time, we use the predicted node scores to search efficiently over the space of interventions. To address the third challenge, We develop a Bayesian formulation that allows to integrate the "counterfactual" information with the score predictions. Next, we describe each component in more detail.

### 4.1 THE *Event Tree*: A TREE OF POSSIBLE FUTURES

We now describe the main data structure that we use to represent the search problem - the *Event Tree*. The event tree is designed to provide a searchable data structure for realizable sequences of events. To make these searches efficient, we represent the system behavior at the *semantic* level. These could be any key interactions between system components, like manufacturing an item in supply chains, or a protein binding the DNA to regulate the expression of a gene. Importantly, in our approach, we require that it is possible to compute the state of the system right after an event.

Each node corresponds to a *sequence* of semantic events. The node's children correspond to realizable continuations of the event sequence. Namely, all possible events that could happen after the sequence $S_u$. We now formally describe the node properties and the expansion of the event tree.

**Tree Nodes.** A node $u$ of the event tree corresponds to the subset of interventions $Y_u \subset \mathcal{Y}$ that share the same *prefix* of semantic events. Each node has a unique prefix $S_u \triangleq (s_1, s_2, \ldots s_u)$. Specifically, after the shared prefix, different sequences of events may follow for different $y \in Y_u$.

The root node describes the set of possible interventions at $t = 0$ and its sequence of events $S_{root}$ is empty. Its *intervention subset* is $Y_{root} = \mathcal{Y}$. See Figure 2, top.

We define $w_y^u$ as the state of the system after it evolved from $y \in Y$ and yielded the sequence $S_u$. Then, a "world-state" of a *node* is defined as the set $W_u = \left\{ w_y^u | y \in Y_u \right\}$.

Given the world state of a current node, we propose *an event-driven forward model* $f(\cdot)$. It takes as input a state $w_y^u$ and outputs the next immediate semantic event. We parallelized it on a GPU to detect possible futures for the world state $W_u$. Appendix I describes the forward model in detail.

**Node Expansion.** Suppose we decide to expand node $u$. We apply the forward model $f(\cdot)$ to each $w_y^u \in W^u$. A new node $u'$ is composed of all $f(w_y^u)$ that share a same next semantic event $s'$. Then, the event sequence of the child node $u'$ is $S_{u'} = \text{concat}(S_u, s')$; the intervention set is $Y_{u'} = \{y | Q(y) \text{ has prefix } S_{u'}\}$.

Expanding the tree can be viewed as a tessellation refinement of the intervention space $\mathcal{Y}$. At each step, we pick one cell and split it into multiple cells, where each child cell represents a different event that occurs after a shared sequence of events, represented by the parent cell.

If the tree is fully expanded, it covers all possible futures. However, expanding the whole tree is expansive, as it grows exponentially with its depth. In the next subsection, we discuss how one can learn a scoring function and use it to guide an efficient tree search.

### 4.2 ASSIGNING AND LEARNING A SCORING FUNCTION FOR NODES

To search the tree for a node that satisfies the goal, we prioritize which node to expand by learning a *scoring* function that assigns scores for nodes, conditioned on the instruction $g$. There are three key challenges in learning a score function. First, we do not have ground-truth (target) scores for tree nodes, and it is unclear what would be an effective assignment of scores. Second, the training data contains only positive examples of correctly designed plans. Finally, we wish to leverage the information about the faulty observed cascade, which is only available during inference time.

To motivate our approach, consider the following naive approach to set target scores. For a given tree, let the "target" $u^*$ be the node that represents the ground-truth sequence $S_{u^*}$. A natural choice for setting scores would be to set $V(u^*) = 1$, and set all other scores to zero. However, this provides little guidance for searching the tree, as no signal is provided until the search hits the target node. Instead, a desired property of the learning algorithm would be to guide the search by assigning monotonically increasing scores along the path from the root to $u^*$.

To address the three challenges we design a principled probabilistic approach for setting the score function. We train our model to predict the likelihood that a sample from $Y_u$, when rolled out, will satisfy the instruction $g$.

$$V(u) = \Pr\left(Q(y) \text{ satisfies } g | y \in Y_u, g\right). \tag{1}$$

Here, nodes on the path from the root to $u^*$ are assigned *monotonically increasing* scores, as the tessellation gets finer and concentrates on $Y_{u^*}$. Additionally, this probabilistic perspective allows us to take a maximum-likelihood approach at inference time to prioritize nodes.

We use the maximum likelihood estimate of $V(u)$ to calculate the ground-truth scores for training. For that, we take a finite sample of $\hat{Y}_{root} \subset \mathcal{Y}$, collecting say $10^6$ points, and use it to expand the tree. The node's score is then the fraction of the samples from $\hat{Y}_u$ that reach the target node $u^*$.

$$\widehat{V}(u) = \Pr\left(y_i \in \hat{Y}_{u^*} | y_i \in \hat{Y}_u, g\right) = \left\|\hat{Y}_{u^*} \cap \hat{Y}_u\right\| / \left\|\hat{Y}_u\right\|. \tag{2}$$

Nodes outside that sequence get a score of 0. In Section 5, we empirically explore alternative approaches for assigning ground-truth scores to nodes.

**Counterfactual update for the score function.** During inference, we observe a cascade that does not satisfy the instruction, and are asked to retrospectively suggest a better solution. How can the observed cascade be used to find a solution? The probabilistic score function we defined allows us to formalize this problem in a Bayesian setting. We treat the model predictions as a *prior* for the true score, and the information about the observed cascade as *evidence*. We then ask how to update the score function given the observed evidence. Formally, our goal is to solve Eq. (1) when it is conditioned by the evidence, $V(u|S_{u^{obs}}$ doesn't satisfy $g)$.

During training, our model learns to estimate the *unconditioned* score function $V(\cdot)$. In the appendix, we show that we can express the Bayesian update of the scores in terms of $V(u^{obs}), V(u)$,

$$V(u|S_{u^{obs}} \text{ doesn't satisfy } g) = V(u) - V(u^{obs}) \cdot fr(y^{obs}, y_u). \tag{3}$$

where $fr(u^{obs}, u) = \Pr(y \in Y_{u^{obs}} | y \in Y_u)$ is the probability that an intervention $y \in Y_u$ will result in sequence with prefix $S_{u^{obs}}$. It is estimated in a fashion similar to Eq. (2).

**A model for the score function.** Next, we describe the representation and architecture for modelling the score function. The model takes as inputs the instruction $g$ and sequence of events $S_u$ that define the node $u$, and predicts a scalar score with ground-truth labels according to Eq. (2).

A naive approach is to represent $S_u$ as a sequence. However, such representation may not convey well the relations describing the cascade of events. For illustration, in the following sequence of collision events *[(A, B), (C, D), (A, E)]*, the collision (A,E) is driven by (A,B), because A is common for both, while (C,D) is less relevant for describing the events that lead to (A,E). Instead, we transform each sequence to a Directed Acyclic Graph (DAG) that captures relations in the cascade of events. A node in this DAG is an event that involves some elements. Each edge represents an element shared by two subsequent events. See Figure 3 for a concrete illustration.

**Architecture** We use a Graph Neural Network (GNN) to parameterize our score function. We represent the graph as a tuple $(A, X, E, z)$ where $A \in \{0,1\}^{n \times n}$ is the graph adjacency matrix, $Y \in \mathbb{R}^{n \times d}$ is a node feature matrix, $E \in \mathbb{R}^{m \times d'}$ is an edge feature matrix, and $z \in \mathbb{R}^{d''}$ is a global graph feature. We chose to use a popular message passing GNN model (Battaglia et al., 2018). We describe its architecture in detail in the appendix.

## 4.3 INFERENCE

Our agent searches the tree for the maximum scored node $u_{MAX}$. Then, it randomly selects an intervention from its intervention subset $y \in Y_{u_{MAX}}$. We consider two variants.

**Maximum likelihood search:** The agent performs a tree search that expands the most likely nodes. At any given step, the agent stores a sorted list of nodes together with their likelihood scores, it then picks the highest scoring node from this list and expands it. The node children are then added to the list with their predicted scores, and the agent resorts the list.

We limit the tree search to expand only 80 nodes, whereas in our test bed a full event tree, which contains all possible realizations, have billions of nodes, $\sim \times 2.8$ per unit of depth (empirically).

**Counterfactual search:** Here we explain how we leverage the information in the *observed* cascade for inference. Consider the case where the sequence of the solution is complex and the observed sequence diverges from the solution at a late point. In this case, it is likely that a part of the observed chain will be informative about the solution, and will diverge at some point. To use that information, we apply the Bayesian correction (Eq. 3) term to the predicted score of every node along the observed sequence. We pick the highest scoring node, and initialize the search up to that node. Then we continue the search as described by the "Maximum likelihood search". In practice, we trim the observed sequence, at $N_{observed}$ nodes. $N_{observed}$ is a hyper-parameter.

## 5 EXPERIMENTS

We compared our approach to state-of-the-art baselines, including human performance. Then, we follow with an ablation study to examine the contributions of different components in our approach. Next, we describe our experimental protocol, compared methods, and evaluation metrics.

### 5.1 A SIMULATION BENCHMARK

We designed a well-controlled environment that share key ingredients with real-world systems of cascading events. [1]  Making it (1) sensitive to initial conditions, to allow a diverse set of future cascades, (2) containing diverse scenes, each describing a unique dynamical system. (3) including semantic goals that depend on intermediate outcomes; and (4) can benchmark counterfactual scenarios.

**Scenes.** In our test-bed, several spheres move freely on a frictionless table, colliding with each other and with static pins within a confined four-walled space (Figure 1). Each episode describes a different scene, which includes tens of collisions. **Instructions.** A structured instruction describes (i) A pivot element to manipulate

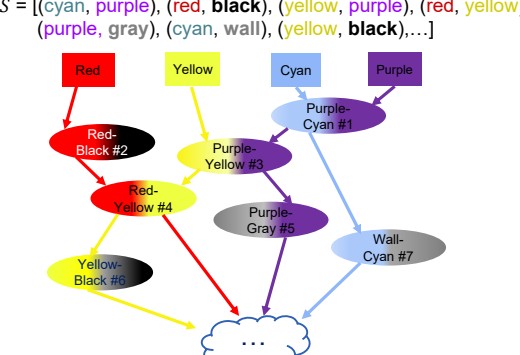

Figure 3: Illustrates transforming a sequence of events (top) to a DAG (bottom). It corresponds to the video in Figure 1 bottom.

"*Push: green ball*"; (ii) A target semantic event (collision) to fulfill "*Target: red hits black pin*"; and (iii) constraints, of two possible types. First, is a "count" constraint. It resembles constraining the total amount of resources available on a logistic chain. It specifies an accumulated number of collisions, on *all* the paths from the pivot to the target, e.g. "*Chain Count: 3*". Second, is a "bottleneck" constraint, which resembles a bottleneck along a logistic chain. It enforces a specific collision that lies on *any* path that starts from the pivot and reaches the target collision, e.g. "*Bottleneck: red hits top wall*". The appendix describes the instruction generation process with more examples. **The task.** The objective of the agent is to intervene with the initial state of the scene, by setting the velocity vector of the pivot object to reach a set of collisions specified by the instruction. This often requires a precise "trick shot", that requires detailed reasoning on how downstream events will roll out.

**Dataset:** We generated a dataset with $\sim 46K$ scenes (we limited generation time to 80 hours), each includes 4-6 moving balls, 0-2 pins, and 4 walls and up to 5 semantic instructions ($\sim 4.25$ on average). The data is split by unique *scenes*, into 470 unseen scenes for test, 69 scenes for selecting hyper-parameters (val. set), and the rest are used for training. See Appendix H for more details.

### 5.2 EXPERIMENT DETAILS

**Compared Methods:** We compared the following methods. **(1) ROSETTE (Reasoning On SEmanTic TreEs)**: Our approach described in Section 4. Search uses the "counterfactual" variant of the tree search (Section 4.3), by first expanding the nodes along the "observed" sequence. **(2) ROSETTE-max-l.**: Like #1, but using "Maximum likelihood search" (Section 4.3) - not using the "observed" sequence. For a fair comparison, we make sure that ROSETTE expands the same number of nodes in total as ROSETTE-max-l. **(3) (Qi et al., 2021)**, The SOTA on PHYRE, using a learned forward model, goal-satisfaction classifier and exhaustive search. For a fair comparison we replace their learned forward model by the full simulator of Makoviychuk et al. (2021). **(4) Cross Entropy**: A standard planner (de Boer et al., 2005; Greenberg et al., 2022) that optimizes the objective function learned by compared method (3). **(5) Sequential**: Using a sequential representation for a tree chain, instead of a DAG. Specifically, we represent the sequence as a graph with edges along the sequence. (Litany et al., 2022) compared a recurrent versus standard synchronous propagation in GNN models and found them empirically equivalent. **(6) Deep Sets regression**: Embedding the instruction and the initial world state to predict a continuous intervention. We embed the objects' initial positions and velocities using the permutation-invariant "Deep Sets" architecture (Zaheer et al., 2017),

---

[1]Examples: link #1, link #2, link #3. Code and data will be released upon publication.

and use an $L_2$ loss with respect to ground-truth interventions in the "counterfactual" training samples. **(7) Random**: Sample interventions at random from an estimated distribution of ground-truth interventions. More details appear in the appendix.

**Ablation:** We also carry a thorough ablation study: First, we explore alternative approaches to label node scores along the ground-truth sequence. *Linear*: Linearly increases the score by $V(u) = depth(u)/depth(u^*)$. *Step*: Give a fixed medium score to nodes along the sequence, and a maximal score to the target node: $V(u) = 0.5 + 0.5\mathbf{1}_{u^*}(u)$. *Dirac-Delta*: Sets $V(u) = \mathbf{1}_{u^*}(u)$, this baseline is equivalent to naive approach discussed in Section 4.2. Second, we compare the "Counterfactual" to the "Maximum Likelihood" search by emphasizing the former effect with **a dataset that employs more complex instructions** using a third constraint, and compare the two types of search across "Easy" and "Hard" instructions. We describe this dataset in the appendix. Last, we test how ablating parts of the instruction affects the ROSETTE model performance. Implementation details of the baselines and ablations are described in Section D.

|  | TREE SUCCESS | SIMULATOR SUCCESS |  | TREE SUCCESS |
|---|---|---|---|---|
| RANDOM | NA | $17.6 \pm 0.3\%$ | DIRAC DELTA | $33.5 \pm 1.6\%$ |
| DEEPSET REGRESSION | NA | $18.4 \pm 0.5\%$ | STEP | $45.1 \pm 1.0\%$ |
| (QI ET AL., 2021) | NA | $21.1 \pm 0.9\%$ | LINEAR | $48.7 \pm 0.7\%$ |
| CROSS ENTROPY | NA | $20.9 \pm 0.4\%$ | ROSETTE-MAX-L (OURS) | $59.7 \pm 0.3\%$ |
| SEQUENTIAL | $52.4 \pm 0.6\%$ | $43.1 \pm 0.3\%$ |  |  |
| ROSETTE (OURS) | $\mathbf{60.8 \pm 0.3\%}$ | $\mathbf{48.8 \pm 0.3\%}$ |  |  |

Table 1: **Success rates. (Left)** Our approach and baselines. TREE is not applicable to the first three baselines since they do not use an event tree. **(Right)** Variants of the score function (see Ablation).

**Evaluation metrics:** For each episode and goal, we predict an intervention and evaluate their success rate using the following metrics. **Simulator success rate**: The success rate when rolling out the predicted intervention using a physical simulator (Makoviychuk et al., 2021). This metric mimics experimenting in the real world. **Tree success rate** (where applicable): Each node in the tree represents a sequence of events. A tree based algorithm selects a node. A "tree success" is when the selected node's sequence satisfy the instruction. This metric evaluates the performance of the score function and tree search, independently from errors that may be introduced due to the event-driven forward model.

We further measured refinements of these metrics by conditioning on various properties of the instruction and scene. **(1) Condition tree success rate on in-**

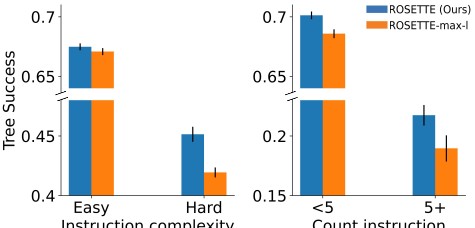

Figure 4: Comparing "Counterfactual" search (ROSETTE) with "Maximum likelihood" search (ROSETTE-max-l) for 2 levels of instruction complexity ("Hard": 2 or more constraints) and for two levels of "count" instructions ("5+": 5 or more ). Using the observed cascade, ROSETTE performs better in complex scenarios.

**struction type: Unconstrained:** The instruction only specifies target collisions. **Bottleneck:** also contains an "bottleneck" constraint. **Count:** contains a "count" constraint. **B&C:** contains both "bottleneck" and "count" constraints. **(2) Condition tree success rate on complex scenarios:** (2.1) Instructions with 2 or more constraints are marked as "Hard", and the rest as "Easy"; (2.2) Instructions with a "count" constraint value $\geq 5$ are considered hard. Complex scenario conditioning was evaluated on the complex instruction dataset. Using the main dataset demonstrate a similar trend (See appendix K). We report mean value and standard error of the mean across 5 model seeds.

## 5.3 HUMAN EVALUATION

To assess a human baseline, we conducted a user study with Amazon Mechanical Turk. We designed a game, where a player is given a video of the observed cascade and is asked to select one of 44 combinations of orientations (11) and speeds (4). The game is based on 30 test episodes. For comparing with ROSETTE, we select the one (of 44) which is nearest (in $L_2$) to ROSETTE's predicted velocity. Appendix Section C, describes the experiment design and further analysis of the results.

## 6 RESULTS

We first compare the performance of ROSETTE with baseline methods and human performance. We then study it properties in greater depth, through a series of ablation experiments. Finally, we discuss the results of the baselines in Appendix A.1, and we provide qualitative examples in Appendix B.

Table 1 (Left) describes the *Tree* and the *Simulator* success rates of ROSETTE and compared methods. ROSETTE achieves the highest success rate for both the "Tree" success rate ($60.8\%$) and the "Simulated" success rate ($48.8\%$). Achieving $\sim 80\%$ conversion rate from *Tree* to *Simulated*. The random baseline success rate is ($17.6\%$), which is close to the performance of the regression model. We conjecture that the regression model fails, because it can't "imagine" the outcomes of the input states as ROSETTE can. The Sequential approach is the strongest baseline, reaching $Tree = 52.4\%$ and $Simulated = 43.1\%$ success rates.

Next, we describe the success rate in the human study. ROSETTE achieves the highest average success rate ($43.3\% \pm 1.3\%$ vs $23.9\% \pm 2.6\%$). Humans displayed a large range of success rates, with the best human achieving $41.4\%$, while the median and worst humans were $25\%$ and $10\%$. ROSETTE performed more persistent, with $46.6\%, 43.3\%$ and $40\%$ for the best, median, and worst.

**Ablation experiments:** **(1)** Table 1 (Right) shows the advantage of the probabilistic formulation of the score function (ROSETTE-max-l), compared to the several heuristics described in Section 5. The strongest baseline ("Linear") only reaches $48.7\%$ vs. $59.7\%$ for ROSETTE-max-l. **(2)** Figure 4 quantifies the benefit gained by using "Counterfactual" search (ROSETTE) over Maximum-Likelihood search (Section 4.3). ROSETTE shows a relative improvement of $7.7\%$ ($45.1\%$ vs $41.9\%$) for complex instructions. **(3)** Table 2 (Appendix A) allows an in-depth examination of the strengths and weaknesses of ROSETTE, across 4 types of ablations, as described in Section 5. First, we observe that the sequential baseline can find target collisions that depend on a bottleneck collision, as well as ROSETTE. However, it fails with "count" instructions ($46.3\%$ vs $60.8\%$), since it has no capacity for that reasoning task. Second, we observe that ROSETTE effectively uses the instruction, since any ablated part of the instruction hurt the respective success rate.

## 7 DISCUSSION

In this paper, we took a first step towards understanding how to affect a complex system of cascading events. We presented a new learning setup, called *Cascade*, where an agent observes a cascade of events in a dynamical system and is asked to intervene and change its initial state to make the system meet a given goal. We use an event-tree representation and a principled probabilistic score function for searching efficiently over the space of interventions. We also describe an approach to counterfactually reason about an observed cascade during the tree search.

Our approach is best applied in problems that are naturally described by event-driven dynamics. As an example, consider cascading failures in power grids (Schäfer et al., 2018). Here, semantic events are failures of nodes (transformers, power generators, ...) or edges (power lines). The power flow obeys a known set of ODE for a given grid. When flow exceeds a powerline capacity, that line fails (an event), resulting in an effectively different grid and a different set of ODEs that govern the dynamics. The transmission system operator may wish to define goals like "no more than three failures", "no more than $n$ people affected", "that important node must not fail". We elaborate on this use-case and other use-cases, such as logistics and evolution of natural disasters in Appendix J.

One important question remains, how do studies that use our toy testbed can generalize to real-world scenarios. We believe it can follow a similar paths as in other areas of AI where approaches mature from toy datasets to realistic problems: First, by creating a benchmark dataset for a real-world domain, annotated with semantic events. Some fields have datasets that can be very natural for the problem we discussed. These may include logistics (Appendix J), evolution of natural disasters and their consequences (Zuccaro et al., 2018), and cascading failures in power grids Schäfer et al. (2018). Second, an event-driven forward model (EDFM) needs to be trained using this dataset. Domain specific properties can be used to improve the accuracy and robustness of an EDFM learned. Finally, given the EDFM, our approach can be applied.

**Reproducibility Statement** We provide full experimental detail about our approach and baselines in Appendix D. Appendix E describes the model architecture, feature represntation, and the data for training the scoring model for the event tree. Appendix H describes the details about the data generation. Appendix I describes the derivation and implementation of the event driven forward model. Finally, Appendix C describes the details about the user study.

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

## A ADDITIONAL RESULTS

Here we describe additional results and provide further discussion.

|  | UNCONSTRAINED | BOTTLENECK | COUNT | B & C |
|---|---|---|---|---|
| ROSETTE (OURS) | $76.5 \pm 0.8\%$ | $68.5 \pm 1.0\%$ | $60.8 \pm 0.7\%$ | $49.5 \pm 0.5\%$ |
| -COUNT | $75.8 \pm 0.9\%$ | $68.4 \pm 0.8\%$ | $6.1 \pm 0.6\%$ | $17.1 \pm 0.5\%$ |
| -BOTTLENECK | $76.4 \pm 0.9\%$ | $21.5 \pm 1.1\%$ | $61.1 \pm 1.2\%$ | $34.1 \pm 0.7\%$ |
| -COUNT -BOTTLENECK | $76.6 \pm 0.8\%$ | $21.4 \pm 1.1\%$ | $6.1 \pm 0.5\%$ | $4.2 \pm 0.3\%$ |
| -FULL | $60.8 \pm 0.5\%$ | $32.7 \pm 0.9\%$ | $11.7 \pm 0.5\%$ | $6.6 \pm 0.1\%$ |
| SEQUENTIAL | $77.9 \pm 0.4\%$ | $66.9 \pm 1.3\%$ | $46.3 \pm 1.3\%$ | $36.5 \pm 1.3\%$ |

Table 2: **Ablation study**. In red, results that perform much worse than ROSETTE.

### A.1 BASELINE RESULTS DISCUSSION

**(Qi et al., 2021)** baseline: We believe that the (Qi et al., 2021) baseline fails for three main reasons: (1) The baseline does not use an event-based representation. (2) It employs a classifier that is trained to provide an all-or-none signal, rather than guiding the search. The ablation study (Table 1, right) demonstrates the importance of guiding the search (compare "Ours" 59.7% vs "Dirac-Delta" 33.5% ) (3) The baseline architecture cannot reason over the temporal DAG structure of a cascade, as our GNN can. The importance of capturing the DAG structure is demonstrated when comparing "Our" (60.8%) to the SEQUENTIAL baseline (52.4%) (Table 1, left)

**Existing Planners:** We wish to provide further insight into why it is challenging to apply existing planners to this setup. The main challenge is that the optimization objective is given in semantic terms about the end goal. To apply a Cross-Entropy-Method (CEM) planner, we derive a corresponding objective function by training a classifier that checks if the goal was achieved for a given scene and plan. Specifically, we used the existing SoTA PHYRE classifier Qi et al. (2021). A main drawback is that classifiers provide an "all or none" signal, hence fail in guiding the planner through optimization. Conversely, our approach provides a score (Eq. 1) that monotonically increases through the tree, and it is constructed to assist the search.

## B QUALITATIVE EXAMPLES

Here we provide links to qualitative examples that we uploaded to YouTube. They are best viewed in $\times 0.25$ slow motion. The YouTube account we use is anonymous.

For each episode, we show a side-to-side video of the observed cascade, the ROSETTE successful case, and ROSETTE-max-l failure case. The instruction is displayed on top of each video.

- *Push: cyan ball, Target: blue hits red, Bottleneck: cyan hits bottom wall, Chain Count: 6*
  In this example, ROSETTE semantically follows the first 10 collisions (in chonological order) as in the observed cascade. It then diverges from the observed cascade, making the blue hit the red. The cyan pivot comes into play already on the 1st collision, and the agent adjusts its velocity such that it shall yield the goal. The ROSETTE-max-l baseline, hits the target, however it fails with the count constraint. The bottleneck collision occurs, but not on the chain from the pivot to the target. See the complete video here: `https://youtu.be/RCKFBRrCRw0`

- *Push: cyan ball, Target: green hits red, Bottleneck: purple hits red, Chain Count: 4*
  In this example, ROSETTE semantically follows the first 5 collisions (in chonological order) as in the observed cascade. The cyan pivot comes into play on the 3rd collision. It then diverges from the observed cascade, and follows another chain of events, making the purple hit the red, and concluding with the target hit within 4 collisions in the chain that started at the cyan ball. This task is too hard for the ROSETTE-max-l baseline, as it completely fails to satisfy the instruction. See the complete video here: `https://youtu.be/4s9MmY2J__I`

- *Push: green ball, Target: green hits cyan, Bottleneck: green hits purple, Chain Count: 5*
  In this example, ROSETTE semantically follows the first 6 collisions (in chonological

order) as in the observed cascade. It then diverges from the observed cascade, making the green hit the purple, fulfilling the bottleneck constraint. The cyan pivot comes into play only on the 6th collision, and the agent adjusts its velocity such that both the target and the count constraint will by satisfied. The ROSETTE-max-l baseline, hits the bottleneck, however it fails to hit the target. See the complete video here: `https://youtu.be/iMedd_7YndQ`

- *Push: yellow ball, Target: cyan hits red, Bottleneck: purple hits red, Chain Count: -*
  In this example, ROSETTE semantically follows the first 5 collisions (in chonological order) as in the observed cascade. The yellow pivot comes into play only on the 5th collision, and the agent adjusts its velocity to satisfy the bottleneck constraint and the target. The ROSETTE-max-l baseline, completely fails in this task. See the complete video here: `https://youtu.be/QLMTD6R2Z54`

- *Push: red ball, Target: blue hits red, Bottleneck: red hits bottom wall, Chain Count: -*
  In this example, ROSETTE semantically follows the first 4 collisions (in chonological order) as in the observed cascade. The red pivot comes into play already on the 2nd collision, and the agent adjusts its velocity to satisfy the bottleneck constraint and the target. The ROSETTE-max-l baseline, satisfy the bottleneck but does not satisfy the target. See the complete video here: `https://youtu.be/vT1ivd1ECJs`

- *Push: yellow ball, Target: blue hits purple, Bottleneck: blue hits right wall, Chain Count: -*
  In this example, ROSETTE semantically follows only the first 2 collisions (in chonological order) as in the observed cascade. The red pivot comes into play on the 3rd collision, and the agent adjusts its velocity to satisfy the bottleneck constraint and the target. The ROSETTE-max-l baseline, completely fails in this task. See the complete video here: `https://youtu.be/rgzWBfx-LqY`

Importantly, these examples demonstrate the usefulness of the observed cascade for tree search. ROSETTE followed the observed cascade along the part of the path that was useful to satisfy the instruction. It diverged from the path when necessary, and found a solution when long cascades were essential, while ROSETTE-max-l struggled.

Finally, we note that this observation is also quantitatively supported: As we show in Figure 4 and Figure A.3. When conditioning the Tree Success rate on producing long cascades, with *Chain count* constraint values greater or equal to 5, ROSETTE performs at $34.8 \pm 0.8\%$, while ROSETTE-max-l performs at $31.3 \pm 1.1\%$, showing ~11.1% improvement. For *Chain count* values smaller than 5, they are statistically equivalent $75.1 \pm 0.4\%$ and $75.4 \pm 0.4\%$.

## C  USER STUDY

We conducted a user study with Amazon Mechanical Turk (AMT) using 30 test episodes. We designed a game where a player (rater) is given a video of the observed cascade and is asked to select one of 44 combinations of orientations (11) and relative speeds (4) (magnitude of velocity). One combination of orientation and speed was aligned with the ground-truth solution, and the rest were spaced in relation to that solution. In an offline stage, we tested which of the *other* orientations and speeds satisfy the goal and included those as valid solutions. We allowed the players to freely replay the observed video. We paid 1$ per game.

Figure A.1 shows one test episode. The upper panel provides an instruction that states the goal of that specific episode. On the left, we provide a set of simple guidelines. The center panel provides the observed (failed) video. The right panel shows the initial frame, overlaid with the set of possible orientations and a set of HTML radio buttons to select the orientation and speed. The upper tab provides a set of four examples with solutions and explanations. Those examples are given in Figure A.2.

To maintain the quality of the queries, we only picked users with AMT "masters" qualification, demonstrating a high degree of approval rate over a wide range of tasks. Furthermore, we also executed a qualification test with a few curated episodes that are very simple. To qualify users, we made sure they do not randomly pick an answer by only qualifying users who completed 5 episodes and had a single error at most. Additionally, we deleted queries from one qualified user,

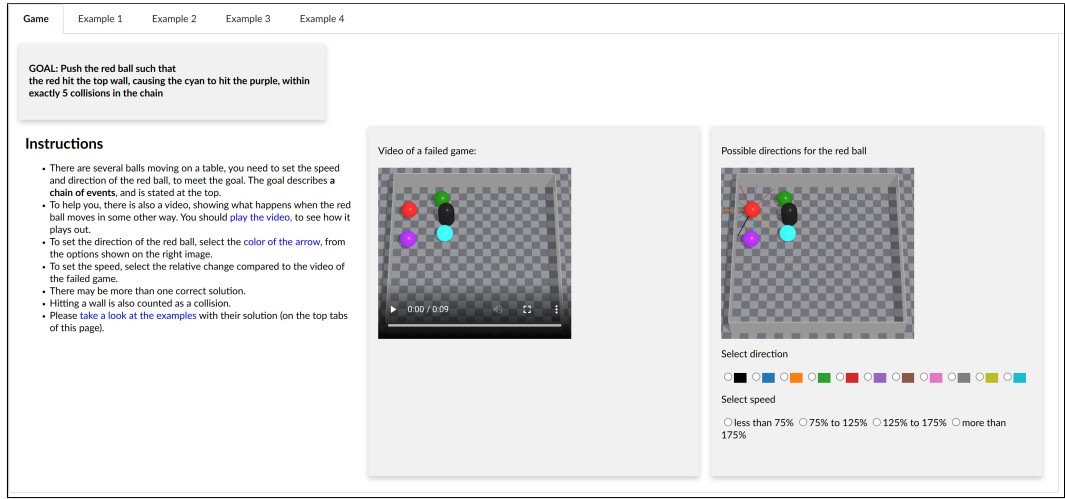

Figure A.1: One test episode of the user study. See Section C for details.

who submitted answers at a rate of 3-4 episodes per minute, as we qualitatively observed that it should take 1-3 minutes to complete an episode.

Qualified users received a bonus of 0.5$, accompanied with the following message:

> Thank you for doing the qualification batch for our colliding balls game.
> Our full study is now online. You can start doing it. Please remember to PLAY
> THE VIDEO and use it to decide about your answer. And also, take another look
> at the examples, as they can provide more intuition about the task.

11 players have passed our qualification tests, playing 25 episodes on average. Table 3 compares the human success rate with ROSETTE and a Random baseline. Showing Average, Median and Best statistics. For the Median and Best statistics, we only included users who played a minimum number of 20 episodes (8 of 11 users).

|  | Average | Median | Best |
|---|---|---|---|
| Random | $17.6 \pm 1.1\%$ | | |
| Humans | $23.9 \pm 2.6\%$ | 25% | 41.4% |
| ROSETTE | **$43.3 \pm 1.3\%$** | **43.3%** | **46.7%** |

Table 3: Success rate statistics for the user study. $\pm$ error denotes the standard error of the mean (S.E.M) across the samples.

# D   ADDITIONAL EXPERIMENTAL DETAILS

## D.1   HYPERPARAMETER TUNING

We train the model and baselines for 15 epochs. Batch size was set to 8192 to maximize the GPU memory usage. We use the PyTorch' default learning rate for Adam (Kingma & Ba, 2015) (0.001). For inference, we set $N_{observed}$ to 9, the maximal tree depth to 30, we sample $10^6$ initial states and expand 80 nodes per episode which takes ~13 seconds. The GNN uses 5 layers, with a hidden state dimension of 128. Hyper parameters were tuned one at a time, during an early experiment on a validation set.

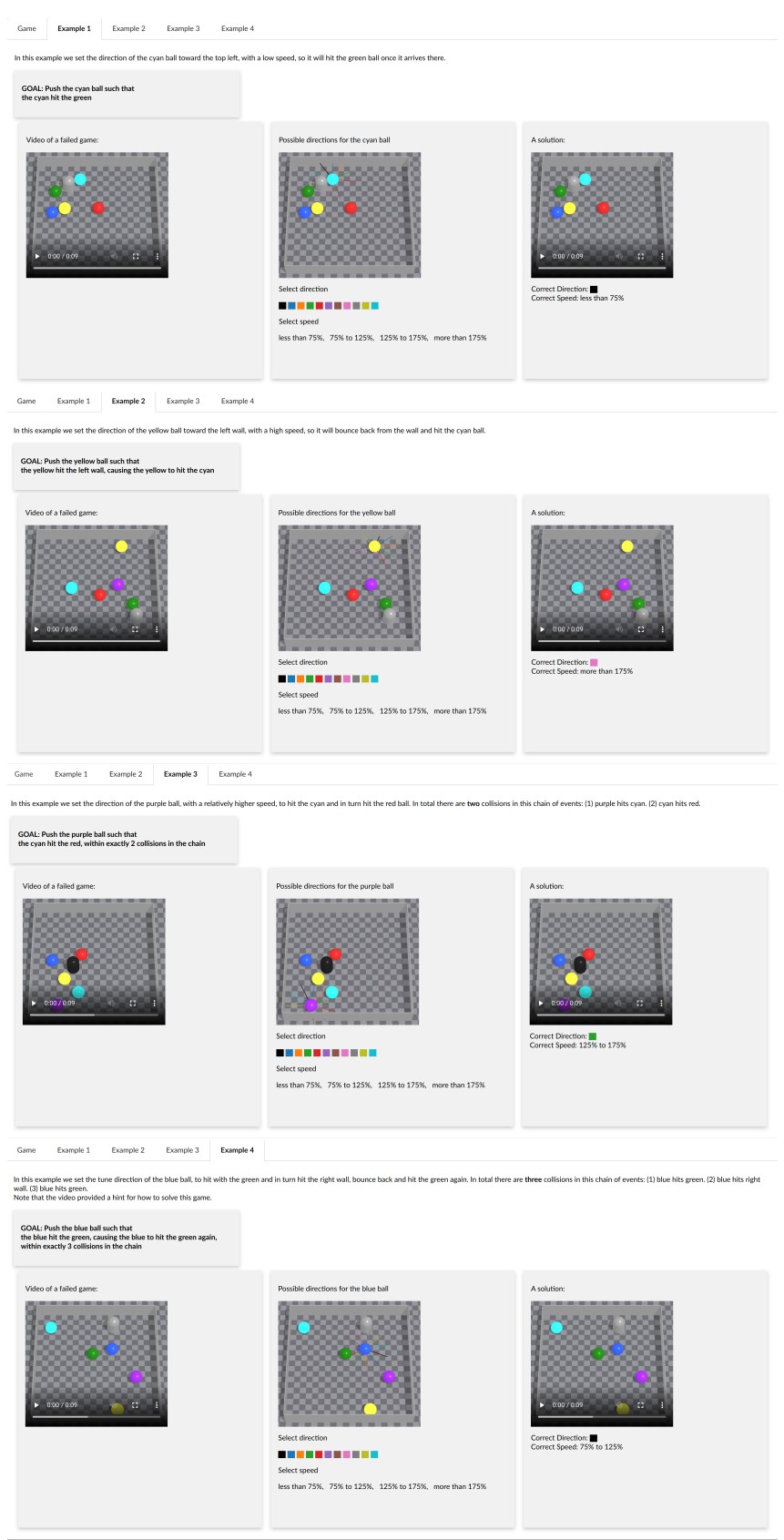

Figure A.2: Examples provided in the user study. See Section C for details.

## D.2 Random Baseline

We sample an intervention at random from an estimated distribution of ground-truth interventions. The distribution is estimated by calculating a 2D-histogram with $30 \times 30$, and approximating the distribution within each bin to be uniform.

## D.3 Deepset regression Baseline

**Overview**: For the Deepset regression baseline, we embed the instruction and the initial world state to predict a continuous intervention. We use the permutation-invariant "Deep Sets" architecture (Zaheer et al., 2017), and use an $L_2$ loss with respect to ground-truth interventions in the "counterfactual" training samples.

The input to the Deepset architecture (Zaheer et al., 2017) is a set of feature vectors. Each feature vector corresponds to a dynamic or static object in the scene. The output is a vector $\in \mathbb{R}^2$, for predicting the controlled velocity of the pivot object.

**Feature representation**: Each feature vector in the set is represented by a concatenation of the following fields $[obj\_feat(o), instruction\_emb, position, velocity]$, where $obj\_feat(o)$ is defined by Eq. (4), $instruction\_emb$, is defined by Eq. (7), $position$, $velocity$ are the initial position and velocity of the object, as given by the observed cascade.

**Labels and loss**: For ground-truth labels, we use the ground-truth velocity of the solution. We use a $L_2$ loss comparing the ground-truth labels with the output of the Deepset architecture.

## D.4 (Qi et al., 2021) Baseline

**Overview**: **Qi2021** is the state-of-the-art approach for solving PHYRE. It uses a learned forward model, a learned goal-satisfaction classifier, and exhaustive search. For a fair comparison with our analytic event-driven forward model, we replace their learned forward model by a full simulator (Makoviychuk et al., 2021).

Therefore, for the goal-satisfaction classifier, in each frame, we replace the set of input feature vectors coming from the region-proposal-interaction-network (RPIN) of (Qi et al., 2021) by a set of feature vectors corresponding to each object in the scene, and its kinematic state as given by the simulator. To condition the classifier on the goal, we concatenate the instruction representation to each feature vector.

**Feature representation**: Each feature vector of an object in a frame, is represented by a concatenation of the following fields $[obj\_feat(o), instruction\_emb, position, velocity, time]$, where $obj\_feat(o)$ is defined by Eq. (4), $instruction\_emb$, is defined by Eq. (7), $position$, $velocity$, $time$ are the respective readings from the simulator in the frame.

**Positive and Negative examples**: For training the goal-satisfaction classifier with positive examples, we use the simulation of the solution cascade. For negative examples, we use the simulation of the observed cascade.

**Classifier Architecture**: We use the classifier architecture of (Qi et al., 2021), as provided in their public implementation, with the following adaptations: (1) We replace the RPIN representation by the simulator-driven representation described above. (2) We allow replacing the last fully connected layer by a multi-layer-perceptron (MLP) (3) We allowed more than four equally spaced input frames.

**Simulator configuration:** The RPIN forward model is a fixed timestamp model, working at 1 frame-per-second. In the full simulator we used (Makoviychuk et al., 2021), we observed that it does not perform well in such a coarse-grained resolution, making objects to sometimes go through the walls. Therefore, we increased the simulator resolution to 10 frames-per-second.

**Hyperparam search:** We searched for the best hyper-parameters configuration that minimizes the validation loss over the following ranges: *Number of MLP hidden-layers* $\in [0, 1, \ldots 6]$, *Number of input frames* $\in [4, 6, 10, 20]$, *batch-size* $\in [128, 256]$. *Number of training epoch* was set according to early stopping on the validation set.

Finally, we used the following hyper-parameters to evaluate the model performance on the test set: *Number of MLP hidden-layers = 2*, *Number of input frames = 4* (as in (Qi et al., 2021) paper), *batch-size=128*, *Number of training epoch = 17*.

**Evaluation:** For evaluation, we randomly selected a subset of 208 episodes (10% of the test subset ), because inference for a single episodes took ~5.5 minutes.

### D.5 CROSS ENTROPY BASELINE

**Overview**: The cross entropy method is a black box optimizer for solving optimization problems. We used (Qi et al., 2021) baseline's classifier as our objective function. At each step, we sampled 100 points and updated the sampling distribution based on their score. We have repeated this process for 100 iterations, and chosen the highest scored intervention for evaluation. Our code is based on a standard implementation Greenberg et al. (2022) of the cross entropy method.

For evaluation, we randomly selected a subset of 208 episodes (10% of the test subset), because inference for a single episodes took ~4 minutes.

### D.6 SEQUENTIAL BASELINE

We used the validation set to select the number of layers for this baseline, $\in [5, 10, 20, 30]$. There wasn't any significant difference when using 5 or 10 layers, and the success rate degraded for 20 or 30 layers. Therefore, we used 5 layers for evaluating performance on the test set.

### D.7 INSTRUCTION ABLATION BASELINES

We report the "count" and "bottleneck" ablations by zeroing their respective features in the instruction and using the same model weights that were used to report the performance of the ROSETTE model. We did not retrain the model for these cases because the ROSETTE model was trained to handle these cases, as is evident by the "Unconstained" metric.

For ablating the "full" instruction, we retrained the model, while completely zeroing the representation vector of the input instruction.

## E IMPLEMENTATION DETAILS OF THE MODEL OF THE SCORE FUNCTION

We use a Graph Neural Network (GNN) to parameterize our score function. We represent the graph as a tuple $(A, X, E, z)$ where $A \in \{0, 1\}^{n \times n}$ is the graph adjacency matrix, $Y \in \mathbb{R}^{n \times d}$ is a node feature matrix, $E \in \mathbb{R}^{m \times d'}$ is an edge feature matrix, and $z \in \mathbb{R}^{d''}$ is a global graph feature. we chose to use a popular message passing GNN model (Battaglia et al., 2018) that maintains learnable node, edge and global graph representations.

**Architecture** The model is composed of several message passing layers, $L^k \circ \cdots \circ L^1$ where each $L^i$ updates all representations, i.e.:

$$X^{i+1}, E^{i+1}, z^{i+1} = L^i(A, X^i, E^i, z^i; \theta^i),$$

Each layer $L_i$ updates the features sequentially: the node and edge features are updated by aggregating local information, while the global feature is updated by aggregating over the whole graph. We denote the parameters of the MLPs that are used in a layer $L_i$ as $\theta_i$, and note that these are the only learnable parameters in the model. At the last layer $i = k$ we use a single dimension for the global feature, i.e., $d' = 1$, which is then used as the score of the event node.

**Feature representation** We describe next the feature representation of the inputs to the node feature matrix $Y$, the edge feature matrix $E$, and the global graph feature $z$.

We start by describing a feature representation of any of the dynamic and static objects in the scene: An object $o$ feature representation, noted by $obj\_feat(o)$, is a concatenation of the following fields

$$obj\_feat(o) = [one\_hot(o), is\_stationary, is\_active,$$
$$instruct\_inner\_prod, bottleneck\_ind, count, count\_ind], \tag{4}$$

where $one\_hot(o)$ is a one-hot vector $\in \mathbb{R}^{12}$, as represented by the instruction; $is\_stationary$ indicates whether the object is stationary; $is\_active$ means that in the context of a current collision, the object dynamics were coming from a collision chain that included the pivot; $instruct\_inner\_prod$ is the results of an inner product of $one\_hot(o)$ with each of the 5 object representations at the instruction embedding (Section H.3). Finally, $bottleneck\_ind, count, count\_ind$ are copied from the instruction embedding.

The graph node and edge features are derived from the DAG representation (Figure 3). Each row of the node feature matrix $Y$ concatenates the two objects that participate at a collision $[obj\_feat(obj_a), obj\_feat(obj_b)]$. Each row at the edge feature matrix $E$ represents $obj\_feat(o)$ of the object on that edge.

Last, the global feature $z$ is a copy of the instruction embedding Eq. (7).

**Training data** For calculating the training labels of the score function, we traverse the semantic tree along the ground-truth sequence of the solution cascade and collect the *positive* labels using Eq. (2). If the event tree cannot reproduce the solution sequence of a sample (due to errors accumulated by the event-driven forward model), then Eq. (2) cannot be calculated, and we drop that sample from the training set. We collect *negative* examples (with $V = 0$) by (1) taking the child nodes that diverge from the path to the ground-truth solution. (2) Traverse a random path along the tree with the same length as the ground truth sequence, and set the score of all the nodes along that path to 0. Note that setting the scores of every node along these paths to $V = 0$ is a heuristic and may introduce some label noise with respect to negative examples. Additional research may be required to analyze the label-noise consequences and address it.

## F    COUNTERFACTUAL UPDATE FOR THE SCORE FUNCTION

In this section, we derive the expression of the score function update according to the observed cascade (Eq. (3)). We start the derivation by repeating the preliminary derivation steps introduced in the main text in more detail.

During inference, we observe a cascade that does not satisfy the instruction, and are asked to retrospectively suggest a better solution. How can the information can be used to find a better solution? The probabilistic score function allows us to formalize this problem in a Bayesian setting. We treat the model predictions as a *prior* for the true score, and the information about the observed cascade as *evidence*. We then ask how to update the score function given the observed evidence. Formally, we condition Eq. (1) by the evidence, $V(u|S_{u^{obs}}$ doesn't satisfy $g)$.

We denote the set of interventions that satisfy the instruction $g$ as $\mathcal{G}_g \subset \mathcal{Y}$, and the evidence by $E$. Note that an equivalent definition for the *unconditioned* score function $V(\cdot)$ is

$$V(u) = \Pr\left(Q(y) \text{ satisfies } g | y \in Y_u, g\right)$$
$$= \Pr\left(y \in \mathcal{G}_g | y \sim U\left(Y_u\right)\right)$$

Our evidence is that for a particular $\tilde{y} \in Y_{obs}$, we have $\tilde{y} \notin \mathcal{G}_g$. Now, by definition, every $y, y' \in Y_{obs}$ share the same observed cascade $S_{u^{obs}}$. Therefore, the evidence $E$ can be equally formulated as $y' \notin \mathcal{G}_g$ for any $y'$ sampled uniformly from $Y_{u^{obs}}, y' \sim U(Y_{u^{obs}})$. For brevity, we set $Y_{obs} = Y_{u^{obs}}$.

The *conditioned* score function is then,

$$\Pr\left(y \in \mathcal{G}_g | y \sim U\left(Y_u\right), E\right)$$

We use the law of total probability and write,

$$\Pr\left(y \in \mathcal{G}_g | y \sim U\left(Y_u\right), E\right)$$
$$= \Pr\left(y \in \mathcal{G}_g y \sim U\left(Y_u\right), E, y \in Y_{obs}\right) \Pr\left(y \in Y_{obs} | y \sim U\left(Y_u\right), E\right)$$
$$+ \Pr\left(y \in \mathcal{G}_g | y \sim U\left(Y_u\right), E, y \in Y_{obs}^c\right) \Pr\left(y \in Y_{obs}^c | y \sim U\left(Y_u\right), E\right)$$
$$= \Pr\left(y \in \mathcal{G}_g | y \sim U\left(Y_u \cap Y_{obs}\right), E\right) \Pr\left(y \in Y_{obs} | y \sim U\left(Y_u\right), E\right)$$
$$+ \Pr\left(y \in \mathcal{G}_g | y \sim U\left(Y_u \cap Y_{obs}^c\right), E\right) \Pr\left(y \in Y_{obs}^c | y \sim U\left(Y_u\right), E\right)$$

Furthermore,

$$\Pr\left(y \in Y_{obs} | y \sim U\left(Y_u\right), E\right) = \Pr\left(y \in Y_{obs} | y \sim U\left(Y_u\right)\right)$$
$$\Pr\left(y \in Y_{obs}^c | y \sim U\left(Y_u\right), E\right) = \Pr\left(y \in Y_{obs}^c | y \sim U\left(Y_u\right)\right)$$

As the conditioned event $y \in Y_{obs}$ is independent of $E$.

The relations between node $U^{obs}$ and $u$ can be one of the three: a) the observed node is a descendant of $u$ (and therefore $Y_u \cap Y_{obs} = Y_u$) b) $u$ and the observed node belong to different branches, and therefore $Y_u \cap Y_{obs} = \emptyset$, or c) $u$ is a descendant of the observed node (and therefore $Y_u \cap Y_{obs} = Y_{obs}$). However, $u^{obs}$ represents a complete cascade rather than a partial sequence, and therefore the observed node does have any children, and we can ignore c).

Let us consider each case separately.

$u$ **and the observed node are along different paths**. In this case,

$$Y_u \cap Y_{obs} = \emptyset$$
$$Y_u \cap Y_{obs}^c = Y_u$$
$$\Pr\left(y \in Y_{obs}^c | y \sim U\left(Y_u\right)\right) = 1$$
$$\Pr\left(y \in Y_{obs} | y \sim U\left(Y_u\right)\right) = 0,$$

and we're left to evaluate $Pr\left(y \in \mathcal{G}_g | y \sim U\left(Y_u\right), E\right)$. Since the evidence in this case provides information about a set that $y$ is not conditioned on, it is independent of $y$, and therefore we conclude with,

$$\Pr\left(y \in \mathcal{G}_g | y \sim U\left(Y_u\right), E\right) = \Pr\left(y \in \mathcal{G}_g | y \sim U\left(Y_u\right)\right) = V(u)$$

$u$ **is a descendant of the observed node.** Here,

$$Y_u \cap Y_{obs} = Y_{obs}$$
$$\Pr\left(y \in Y_{obs} | y \sim U\left(Y_u\right)\right) = fr(y_{obs}, y_u)$$

In this case,

$$\Pr\left(y \in \mathcal{G}_g | y \sim U\left(Y_u\right), E\right)$$
$$= \Pr\left(y \in \mathcal{G}_g | y \sim U\left(Y_u \cap Y_{obs}\right), E\right) fr(y_{obs}, y_u)$$
$$+ \Pr\left(y \in \mathcal{G}_g | y \sim U\left(Y_u \cap Y_{obs}^c\right), E\right) \left(1 - fr(y_{obs}, y_u)\right)$$
$$= \Pr\left(y \in \mathcal{G}_g | y \sim U\left(Y_{obs}\right), E\right) fr(y_{obs}, y_u)$$
$$+ \Pr\left(y \in \mathcal{G}_g | y \sim U\left(Y_u \cap Y_{obs}^c\right), E\right) \left(1 - fr(y_{obs}, y_u)\right) \tag{5}$$

Now,

$$\Pr\left(y \in \mathcal{G}_g | y \sim U\left(Y_{obs}\right), E\right)$$
$$= \Pr\left(y \in \mathcal{G}_g | y \sim U\left(Y_{obs}\right), \{\forall y' \in Y_{obs}, y' \notin \mathcal{G}_g\}\right)$$
$$= 0$$

Since the evidence indicates that for every $y' \in Y_{obs}$ the resulting sequence $S_{u^{obs}}$ does not satisfy the goal. Furthermore,

$$\Pr\left(y \in \mathcal{G}_g | y \sim U\left(Y_u \cap Y_{obs}^c\right), E\right) = \Pr\left(y \in \mathcal{G}_g | y \sim U\left(Y_u \cap Y_{obs}^c\right)\right) \tag{6}$$

As $E$ does not add information when we sample from $(Y_u \cap Y_{obs}^c)$.

Therefore,

$$\Pr\left(y \in \mathcal{G}_g | y \sim U\left(Y_u\right), E\right) = \Pr\left(y \in \mathcal{G}_g | y \sim U\left(Y_u \cap Y_{obs}^c\right)\right)\left(1 - fr(y_{obs}, y_u)\right)$$

Now

$$\Pr\left(y \in \mathcal{G}_g | y \sim U\left(Y_u\right)\right) = \Pr\left(y \in \mathcal{G}_g | y \sim U\left(Y_u \cap Y_{obs}\right)\right) fr(y_{obs}, y_u) \\ + \Pr\left(y \in \mathcal{G}_g | y \sim U\left(Y_u \cap Y_{obs}^c\right)\right)\left(1 - fr(y_{obs}, y_u)\right)$$

Namely,

$$V(u) = V(u_{obs}) \cdot fr(y_{obs}, y_u) + \Pr\left(y \in \mathcal{G}_g | y \sim U\left(Y_u \cap Y_{obs}^c\right)\right)\left(1 - fr(y_{obs}, y_u)\right)$$

or

$$\Pr\left(y \in \mathcal{G}_g | y \sim U\left(Y_u \cap Y_{obs}^c\right)\right) = \frac{V(u) - V(u_{obs}) \cdot fr(y_{obs}, y_u)}{1 - fr(y_{obs}, y_u)}.$$

Plugging this back to Eq. 6 we obtain:

$$\Pr\left(y \in \mathcal{G}_g | y \sim U\left(Y_{obs}\right), E\right) = V(u) - V(u_{obs}) \cdot fr(y_{obs}, y_u)$$

which is our final result.

## G  RELATION TO CAUSAL-INFERENCE

The DAG representation (Section 4.2) is useful for graphically representing one instance of a cascade, but we intentionally avoid naming it a *Causal* DAG, because it can't represent dependencies between events that are not explicitly observed in the video. E.g., in the example *[(A, B), (C, D), (A, E)]* in Section 4.2, it may be that (A,E) depends on (C,D) because C blocks D from reaching to E before A do. The event tree can simulate this behaviour, while the DAG (C,D) ; (A,B)-¿(A,E) is unaware of it. From a formal causal inference perspective (Pearl, 2000), our event tree is the part of our approach that can be related to the formal "Structured" Causal Model (SCM). As it is a generative model that reflects the data generation process; it can account for complex dependencies between events; and every edge corresponds to a function, namely, the event-driven forward model.

## H  DATA GENERATION DETAILS

### H.1  VIDEO GENERATION DETAILS

In this section, we describe the generation process of the dynamical scene. We first create an "unperturbed" video. Then, we perturb the video by modifying the velocity of a specific element, which will be later designated as the pivot. We let the perturbed video roll out, validate that it is indeed semantically different than the unperturbed video, and label it as the "observed" video. The unperturbed video can now be used as reference for our instruction generation process. It is a realization of a specific, complex, semantic chain of events that is both semantically different than the perturbed ("observed") video and is also feasible, e.g, by setting the intervention value as to revert the perturbation. This flow guarantees that we can ask meaningful instructions on the "observed" that are guaranteed to be realizable.

**The unperturbed video.** We construct the unperturbed video by iteratively adding spheres and collisions in a physical simulator (IsaacGym (Makoviychuk et al., 2021)) increasing the video complexity. We start by placing a sphere in the confined four-walled space and assign it a random velocity.

The dynamics of a sphere moving freely in a confined square area can be expressed analytically. We pick a random time $t_1$, hitting velocity, and hitting angle for the first collision. We analytically solve for the initial position and velocity at $t_0 = 0$ that will result in the a collision at $t_1$ with the specific hitting velocity and angle. We assign these value to a randomly colored sphere.

Due to discrepancies between the simulator dynamics and the kinematic analytic model, we roll out the dynamical system in the simulator, and record the system state immediately after a collision.

We continue adding spheres iteratively. Given a state at $t_i$, we randomly select a sphere $O_i$ from the existing spheres, collision time $t_{i+1}$, hitting angle and velocity. We solve analytically and find the initial position and velocity at $t_0 = 0$ that will result in a collision with $O_i$ corresponding parameters. We roll out the dynamical system, and update the velocities and positions records after each collision with the empirical values from the simulator.

Our simple kinematic model assumes the target sphere and the newly added move freely. However, other spheres may cross their trajectories, resulting in an a collision that will distract the spheres from their designated path. However, this simply means that the planned random collision was replaced by a different collision. Since we update our records of the resulting collisions and corresponding output velocities and positions using the simulator, this does not pose any serious limitations.

**The observed video.** We pick a random sphere from the set of spheres and assign it a different velocity at $t = 0$. We roll out the system in the simulator and log all resulting collisions. We validate that the resulting collision sequence is different than the unperturbed video collision sequence. We now have two videos that differ only in the initial velocity of a specific sphere, but result in a substantially different semantic chain of events.

## H.2 INSTRUCTION GENERATION DETAILS

We describe the instruction generation process when given an "observed" video, and a "counterfactual" video that displays an alternative cascade of events.

Given a ground-truth video, its sequence of collisions, and a pivot, we randomly sample an instruction: Starting by randomly sampling a target collision from the sequence. And then, we randomly sample up to two constraints that accompany the goal. For constructing the constraints, we first represent the sequence of collisions using a DAG, in a similar fashion as described in Figure 3, then we use standard NetworkX functionality (Hagberg et al., 2008) for graph traversal: (1) We use "dag.ancestors()" to get a list of nodes for the "bottleneck" constraint. (2) We use "all_simple_paths()" to count the nodes in a chain reaction between the pivot and the target collision.

To avoid trivial goals, we drop an instruction if it is fulfilled by the observed video (rather than the "counterfactual" video). We sample up to $5$ unique instructions for each scene (~4 on average).

## H.3 INSTRUCTION FEATURE REPRESENTATION

We assume perfect lexical perception, and provide the agent with the a structured vector representation of each instruction, by concatenating the following fields:

$$instruction\_emb = [target\_obj\_a, target\_obj\_b, pivot\_obj, bottleneck\_obj\_a,$$
$$bottleneck\_obj\_b, bottleneck\_ind, count, count\_ind], \qquad (7)$$

where $target\_obj\_a, target\_obj\_b$ are the object representations of the target collision. $pivot\_obj$ represents the pivot. $bottleneck\_obj\_a, bottleneck\_obj\_b, bottleneck\_ind$ represent the two "bottleneck" objects and a binary indicator scalar. If an "bottleneck" constraint is not applicable for an instruction, we them all to 0. $count, count\_ind$ are 2 scalar values: One for the number of collisions of the chain "count" constraint, and another used as a binary indicator for the "count" constraint. Similarly if a "count" constraint is not applicable for an instruction, we set both $count$ and $count\_ind$ to 0.

Finally, note that each object is represented by a one-hot vector $\in \mathbb{R}^{12}$, because the environment has 12 types of unique objects: 6 colored balls, 2 static pins, and 4 walls.

### H.4 COMPLEX INSTRUCTIONS DATASET

For the complex instructions dataset, we add a third object centric constraint that counts the number of interactions a specific object makes on the paths from the pivot to the target collision. It resembles constraining the amount of resources available per instance on a logistic chain. With an additional constraint we can test our approach on a more challenging task that has a large variety of instructions that have 2 or more constraints. We split the evaluation set to "Hard" instructions that have 2 or more constraints, and "Easy" instruction with 0-1 constraints. We generated instructions for the same scenes as in the main dataset, which yields $\sim 4.5$ instructions per scene. The test set consists of 2190 episodes, where $54\%$ are "Hard" instructions.

## I THE FORWARD MODEL

In our physical setup, the dynamics are prescribed by the position and velocity $c_i^j = (pos_i^j, vel_i^j), j = 1 \ldots n$ of each of $n$ objects in the environment. The world state $w_i^u$ of a node $u$ is then a tuple

$$w_i^u = (c_i^1, c_i^2, \ldots c_i^n, t_i), \tag{8}$$

where for the root node $t_i = 0$ for all $x_i$.

The forward module takes as input a world state $w_i$ it outputs the next semantic event $(s')$, and a state $f(w_i) = w_i' = (c'_i^1, c'_i^2, \ldots c'_i^n, t'_i)$ immediately after the predicted semantic event at $t'_i$. The section is divided into three parts. First, we describe the analytical equations that control if two objects will collide. Then we show how can leverage the analytic model to efficiently branch out from a node in the event tree. Finally, we fill in the missing details and present the full forward model.

**The collision detector**. Assume two spheres $i = \alpha, \beta$ moving freely on a plane with an initial velocity of $\mathbf{v_i}$ and position $\mathbf{r_i}$ at $t = 0$. Each sphere has a radius of $l_i$. If the two sphere collide, then, at the moment of collision, the spheres intersect at a single point. We can use a simple geometric calculation to find their *planar* distance. The distance between the center of spheres is $l_\alpha + l_\beta$, while the vertical distance between the two centers is $|l_\alpha - l_\beta|$. The resulting planar distance is then:

$$d = \sqrt{(l_\alpha + l_\beta)^2 - (l_\alpha - l_\beta)^2} = 2\sqrt{l_\alpha l_\beta}. \tag{9}$$

Therefore, in order to check if the spheres collide, we can check if the planar distance between the two spheres is ever equal to $d$,

$$\|\mathbf{r}(t)\|^2 = \|\mathbf{r_\alpha} + \mathbf{v_\alpha} \cdot t - \mathbf{r_\beta} - \mathbf{v_\beta} \cdot t\|^2 = d^2 \tag{10}$$

This is a quadratic equation in $t$, which we can solve for analytically. If the discriminant is non-negative, the collision time corresponds to the smaller root. The spheres' velocities immediately after the collision are given by:

$$\mathbf{v_1'} = \mathbf{v_1} - \frac{2m_2}{m_1 + m_2} \frac{\langle \mathbf{v_1} - \mathbf{v_2}, \mathbf{y_1} - \mathbf{y_2} \rangle}{\|\mathbf{y_1} - \mathbf{y_2}\|^2} \cdot (\mathbf{y_1} - \mathbf{y_2}) \tag{11}$$

$$\mathbf{v_2'} = \mathbf{v_2} - \frac{2m_2}{m_1 + m_2} \frac{\langle \mathbf{v_1} - \mathbf{v_2}, \mathbf{y_1} - \mathbf{y_2} \rangle}{\|\mathbf{y_1} - \mathbf{y_2}\|^2} \cdot (\mathbf{y_2} - \mathbf{y_1}) \tag{12}$$

Likewise, it is trivial to obtain an analytical expression for the collision time and output velocity of a collision between a freely moving sphere and each of the static walls bounding the spheres (should the collision occur). The sphere's velocity in the direction orthogonal to the walls flips, while the parallel velocity remains the same.

**Parallelizing collision detection.** The collision detector provides an analytic condition that validates whether a specific collision occurs.

$$(\Delta\mathbf{r} \cdot \Delta\mathbf{v})^2 - 4(\|\Delta\mathbf{r}\|^2 - d^2)\|\Delta\mathbf{v}\|^2 > 0 \tag{13}$$

Eqs. 9-13 can be solved in parallel for multiple tuples of $(\mathbf{r_1}, \mathbf{v_1}, \mathbf{r_2}, \mathbf{v_2})$ on a GPU using packages such as PyTorch. Given an intervention set of $Y_u$, and a corresponding world-state set $W_u$, we iterate over all possible collisions $S_{ij} = (O_i, O_j)$. For each collision between object $i$ and $j$ we can apply our collision detector by extracting the corresponding coordinates $c_w^i, c_w^j, t_w$ from $w \in W_u$ (Eq 8). We can do in parallel for all world states $w \in W_u$. If a collision is predicted, we construct a new node child $u'$ of $u$. We associate with it the interventions for which the collision detector returned a non-null time for the collision, $Y_u'$, the corresponding set of post-collision world state $W_u'$, and the event sequence $S_u' = concat(S_u, (O_i, O_j))$

The complexity is quadratic in the number of object rather than linear in the number of interventions. This allows us to apply our algorithm with a high number of interventions, and therefore enable us to consider delicate sequences of collision that would require refined "trick shots".

This approach considers every two objects $O_i, O_j$ as moving freely. However, another object in the environment, e.g, $O_k$, may interact with $O_i$ (without loss of generality) before the collision. This necessarily means that the collision time $t_{ik}$ precedes $t_{ij}$. In order to account for this, we hold an additional structure that maintains the minimal collision time for every $w \in W_u$. We update it as we iterate over all possible collisions. Then, we associate each $w \in W_u$ and its corresponding $y \in Y_u$ to the event node corresponding to the collision with the earliest collision time.

## J    ADDITIONAL SETUPS

Here, we present examples for additional setups for which our formalism can be applied.

### J.1    LOGISITICS

While logistics is a complex field, we describe a simple model that captures the essential components of a logistics problem.

Consider a large logistics enterprise that needs to coordinate shipping from multiple locations. The enterprise has multiple carriers (e.g, trucks or airplanes) $v_i, i = 1..m$ and routes them between logistic centers at $r_j, j = 1..n$.

A plan is a schedule for each carrier $\tau_j$, where a schedule $\tau_j$ is a sequence of arrivals and departures between various logistic centers,

$$\tau_j = \{(r_j^0, t_{in}^0, t_{out}^0), (r_j^1, t_{in}^1, t_{out}^1), ...\}$$

Not all plans are *feasible*. Each carrier can travel at a range of velocities, resulting in a range of arrival times to the possible destinations. Carriers can exchange cargo is they are present at the same logistic center.

Now, assume a logistic center is suddenly shut down. Rescheduling all carriers is unfeasible, as some may be already committed to a route, or may not be easily diverted (e.g., are airborne). Furthermore, recomputing a new plan for the complete enterprise might be computationally heavy. Finally, it seems reasonable that re-planning of only the routes of carriers that were suppose to arrive to the closed logistic center may be enough. We denote those $k$ carriers as the *rescheduled carriers*. Note that while only some of routes may be re-planned, other carriers might be affected as well due to a cascade of delays or even cargo exchange cancellations.

Such re-planning may be constrained by semantic instruction. For example: "Carrier X should only make two deliveries", "Carrier Y should meet carrier Z before meeting Carrier W", etc. .

We now cast this problem into our general framework, described in Section 4. An event is the arrival or departure of a carrier to a logistic center. Each intervention $y \in \mathcal{Y}$ is a set of plans the rescheduled

carriers,

$$y = (\tau_0, \tau_1, ..., \tau_k).$$

A world state $w_j$ is the position of the $k$ carriers at different time $((p_j^0, t_j^0), ..., (p_j^k, t_j^k))$ The forward model takes as input a world state and outputs all world state that obey the following two rules: 1) At least one carrier moved to a different logistic center. 2) The transition of carriers follow physical constraints. If carrier $i$ can move at velocity range $[v_{min}^i, v_{max}^i]$ and it moves between two logistics centers $r_a$ and $r_b$, then the transition time must be in

$$\left[ \frac{\|r_a - r_b\|}{v_{max}^i}, \frac{\|r_a - r_b\|}{v_{min}^i} \right]. \tag{14}$$

For simplicity, we assume that there is no cargo limit.

The expressions for the induced probability, Eqs. 2 -3, remains the same.

## J.2 FAILURE CASCADES IN POWER GRIDS

Cascading failures in power grid may cause large blackout with substantial economical damage Schäfer et al. (2018). Cascading Power failures may be induced due to random fluctuations and can develop on orders of seconds. Human operators or complex control mechanism may not be able react in time. The transmission system operator may use an event-driven forward model to find fast automated reactions for unseen dynamical configurations to avoid cascading failures.

Here, semantic events are failures of nodes (nodes; e.g., transformers, power generators, etc.) or power lines (edges). Power flow follows a known set of ODE for a given grid (eqs 14-15 in Schäfer et al. (2018)):

$$\frac{\mathrm{d}}{\mathrm{d}t}\theta_i = \omega_i, \tag{15}$$

$$I_i \frac{\mathrm{d}}{\mathrm{d}t}\omega_i = P_i - \gamma_i \omega_i + \sum_{j=1}^{N} K_{ij} \sin\left(\theta_j - \theta_i\right), \tag{16}$$

where, $\theta_i, \omega_i$ are the dynamical variable at node $i$, $P_i$ is the power input (or output) at node $i$, and $K_{ij}$ is a weighted adjacency matrix representing the grid connectivity. If at some point in time the flow $F_i j$ exceeds the powerline capacity $\alpha k_{ij}, \alpha \in [0, 1]$ (eqs 1-2), the line fails. This condition can be formally written as

$$F_{ij}\left(t\right) = K_{ij}\sin\left(\theta_j\left(t\right) - \theta_i\left(t\right)\right) > \alpha K_{ij}.$$

If the line fails, the dynamics are governed by a new effective coupling matrix $K_i j$, and the dynamics in Eqs. 15-16 changes accordingly.

A failure of a node may induce outage to some region. The transmission system operator (TSO) might define goals such as "no more than three failures", "the maximal number of affected people should be less than $n$", "these highly important nodes should not fail" etc.

## J.3 EVOLUTION OF NATURAL DISASTERS

Finally, another use case is the evolution of natural disasters. Zuccaro et al. (2018) provides a full description of an event tree. It models transitions between events like a "seismic shock" which can lead to "landslide" and result with "traffic accident", and how taking preventive measures like "evacuate population" can influence the total damage caused by the crisis.

## K COMPLEX SCENARIO CONDITIONING FOR THE MAIN DATASET

In Figure A.3 we provide the results for complex scenario conditioning for the main dataset (with two type of constraints). The results demonstrate a similar trend as in the complex instruction dataset in Figure 4.

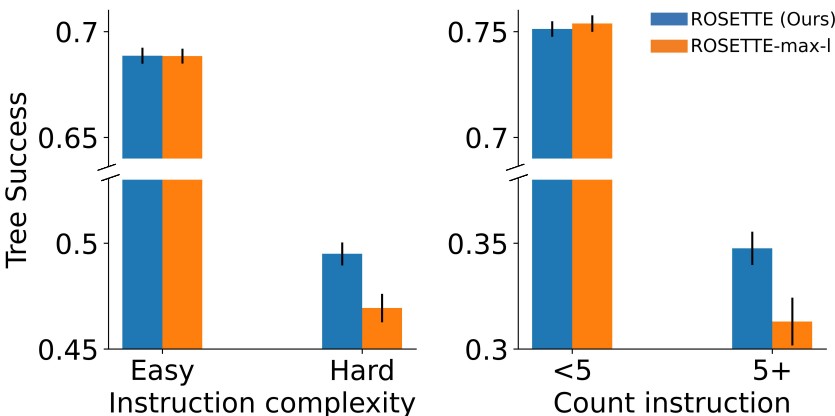

Figure A.3: Comparing "Counterfactual" search (ROSETTE) with "Maximum likelihood" search (ROSETTE-max-l) for 2 levels of instruction complexity ("Hard": 2 or more constraints) and for two levels of "count" instructions ("5+": 5 or more ). **Here we use the main dataset.** Using the observed cascade, ROSETTE performs better in complex scenarios.

