# OpenReview forum: "Learning to Reason and Act in Cascading Processes"
_ICLR.cc/2023/Conference — Submitted to ICLR 2023_

### Official Review · Reviewer_1vrT · 2022-10-25

**Confidence:** 3
**Correctness:** 2
**Technical Novelty And Significance:** 4
**Empirical Novelty And Significance:** 2
**Recommendation:** 3

**Clarity, Quality, Novelty And Reproducibility:**

- Clarity and Quality: The event tree and the proposed planner are somewhat difficult to understand from the writing.
- Novely: Yes, the work is novel.
- Reproducibility: Authors promise to release the data and the code upon publication.

**Strength And Weaknesses:**

### Pros

The idea of an event-based state space, an event-based world model and the idea of doing planning inside it is very interesting.

### Weaknesses/Questions

1. Why not use an existing benchmark like PHYRE/TOOLS if the focus is on proposing a new planning framework? Are event trees not applicable to PHYRE/TOOLS benchmarks? If not, then perhaps the proposed approach is too tailored to the proposed benchmark.
2. Could we apply a standard planner like CEM on event-trees?
3. Very little information about the forward model is provided. Can event-based world models be learned directly from pixels similarly to how we learn latent-variable temporal generative models e.g. RSSM [1]?

[1] Hafner, Danijar et al. “Learning Latent Dynamics for Planning from Pixels.”

**Summary Of The Paper:**

The paper proposes a

1. A new learning setup called Cascade in which the agent is provided a semantic goal and a failed trajectory and the goal of the agent is to predict an intervention for the initial state of a pivot object so that the semantic goal is achieved counterfactually.
2. A new dynamics model based on semantic events.
3. A method to efficiently search or plan using the event-based dynamics model.
    1. A probabilistic scoring function
    2. Leveraging observed cascade to guide search.

**Summary Of The Review:**

The general idea of an event-based world model and the idea of planning in it is very interesting. But I feel that the paper is taking too many steps simultaneously and thus making it difficult to strongly claim any one contribution. For instance, it is hard to judge the value of event trees if the evaluation is done on a new benchmark also proposed in the same paper. Similarly, it is hard to judge the value of event trees if the planning is done by a newly proposed planning algorithm rather than some standard and well-known planner.

---

> ### Author Response · Authors · 2022-11-14
> **Response to reviewer 1vrT (part 1/2)**
>
> Thank you for your valuable feedback and for your time! We completely agree that acting in an event-based world-model is very interesting, and we were encouraged that you found this work to contain “significant technical novelty”. We hope that our response below addresses all your concerns.
>
> The main concern seems to be that this paper introduces a new benchmark and a new method, and we want to be careful not to jointly overfit the method and benchmark without a broader applicability. We address this concern in three ways: (1) We test the previous SoTA PHYRE approach (Qi et al. 2021) on the CASCADE problem. (2) We now added comparisons with a standard planner. (3) In our general response to reviewers we discuss the applicability of the approach to real-world problems that are naturally captured with event-driven models, see also our update to Section 7 (Discussion).
>
> > **Why not use an existing benchmark like PHYRE/TOOLS if the focus is on proposing a new planning framework? Are event trees not applicable to PHYRE/TOOLS benchmarks?**
>
> Event-trees can be applicable to datasets like PHYRE/TOOLS, but that would require access to an event-driven forward model (EDFM), which is currently not available in PHYRE/TOOLS. Training such a model requires additional annotations. Specifically, annotation with semantic events like “box hits pole” or “pole lifts ball”, at specific points in time and space.
>
> So why didn’t we use PHYRE/TOOLS? Because the focus of this work is on *learning to search* given an EDFM model and *not on learning the EDFM model* itself. As the reviewer pointed out this work already has many components and we believe that learning an EDFM model should be studied in a separate work. Several approaches have been proposed for learning event-driven FMs from temporal data, like dynamic Bayes nets (Gharamani 1997, Gunawardana 2016, Bhattacharjya 2020), which can also handle latent variables.
> We now discuss these points in Section 2 (related work).
>
> > **It is hard to judge the value of event trees if the evaluation is done on a new benchmark also proposed in the same paper.  Similarly, it is hard to judge the value of event trees if the planning is done by a newly proposed planning algorithm rather than some standard and well-known planner.**
>
> We agree with the concern that when adding a benchmark and a method, they should be compared with existing approaches. There are usually two ways to do this: (1) evaluate a new method on previous benchmarks (2) Evaluate previous methods on the new benchmark. We did the latter.
>
> Specifically, we evaluated the SoTA approach of PHYRE on our CASCADE benchmark. We find that our approach wins by a large margin (48.8% success vs 21.1%). Our insight is that (Qi 2021) baseline fails for three main reasons: (1) It employs a classifier that is trained to provide an all-or-none signal, rather than guiding the search. (2) It does not use an event-based representation. (3) It cannot reason over the temporal DAG structure of a cascade. We now added a discussion about this baseline results in Appendix A.1
>
> Following the reviewer's suggestion, we also applied a standard Cross-Entropy-Method (CEM) planner as a baseline. It reached a success rate of 20.9% +- 0.44% (S.E.M), compared with our result of 48.8%+-0.3% and similar to the Qi et al baseline (21.1% +- 0.9%). We added this baseline to the paper in Section 5, and its implementation details are described in Appendix D.5. The results are discussed in Appendix A.1, and below.
>
> We wish to provide further insight into why it is challenging to apply existing planners to this setup.
> The main challenge is that the optimization objective is given in semantic terms about the end goal. To apply a CEM planner, we derive a corresponding objective function by training a classifier that checks if the goal was achieved for a given scene and plan. Specifically, we used the existing SoTA PHYRE classifier (Qi 2021). A main drawback is that classifiers provide an “all or none” signal, hence failing in guiding the planner through optimization. Conversely, our approach provides a score (Eq 1) that monotonically increases through the tree, and it is constructed to assist the search.
>
>
> > **Could we apply a standard planner like MPC on event-trees?**
>
> Regarding MPC. That would not be the best match for the problem here because we have a single point of intervention rather than sequential control.
>
>
> **References:**
>
> Qi et al. Learning Long-term Visual Dynamics with Region Proposal Interaction Networks, ICLR 2021
>
> Bhattacharjya et al. Event-driven continuous time bayesian networks, AAAI 2020
>
> Gunawardana et al. Universal Models of Multivariate Temporal Point Processes, AISTATS 2016
>
> Gharamani et al. Learning dynamic Bayesian networks, 1998

---

> > ### Author Response · Authors · 2022-11-14
> > **Response to reviewer 1vrT (part 2/2)**
> >
> >
> >
> >
> > > **Very little information about the forward model is provided.**
> >
> > The event-driven forward model is described in Appendix I. Following the reviewer's comment we updated the paper to elaborate in better detail about (1) its derivation, (2) how it is parallelized efficiently on a GPU, and (3) the connection between the forward model to expanding a new event node in the tree.
> >
> > > **Can event-based world models be learned directly from pixels similarly to how we learn latent-variable temporal generative models e.g. RSSM [1]?**
> >
> > Yes, this is a promising suggestion. We believe that lower-dimension latent-variable representations resonate well with our approach. In fact, the forward model in this paper uses a 4-dimensional representation of velocity and position to represent each object, rather than a pixel-based representation. We now cite RSSM in the related work.

---

> > > ### Comment · Reviewer_1vrT · 2022-12-07
> > > **Reply**
> > >
> > > I thank the authors for the response.
> > >
> > > > We test with Qi et al. and CEM
> > > >
> > >
> > > I still feel that the proposed approach can do better over these baselines mainly because it is the only model that can accept or work explicitly with the event information that their simulator and the semantic constraint provide — suggesting that the approach is too hand-tailored to the proposed benchmark. The proposed model also seems to have access to the (almost) true world model in which obtaining the solution becomes just an exhaustive symbolic search. Given the true model anyway, with enough algorithm-focused engineering, solving this search should not be that difficult. I am still not sure whether the paper indeed provides interesting new lessons for the learning/representation community.
> > >
> > > > Focus of this work is on *learning to search* given an EDFM model. We don’t want an all-or-none signal about a plan’s success.
> > > >
> > >
> > > While the authors say that the paper is about search, in reality, they do propose a new notion of semantic events and event-driven models (EDFMs). If so, then the question of proposing the event-ness comes before the question of searching in it. I am not saying that the search part cannot be a contribution, but it is a bit odd that the search part is the focus when the notion of EDFM is already new. Shouldn't we want to first establish the value of event-drivenness in the learning community before raising the question of search?
> > >
> > > I still maintain my score. The best I can do is reduce the confidence of my rating to let other reviewers take precedence who probably better see the value that I am unable to see.

---

> > > > ### Author Response · Authors · 2022-12-09
> > > > **Reply**
> > > >
> > > > We appreciate the reassessment of your confidence score.
> > > >
> > > > The paper presents a new problem and shows that state-of-the-art algorithms struggle with that problem. Our new method exploits structure in the problem that was not used by other algorithms. The way to establish the value of a structure is by showing how to benefit from it, which is what we do.
> > > >
> > > >
> > > > *Re “the value of event-drivenness”, and the “question of search”:* We established both. The EDFM alone [1] improves the success rate to 33.5% from 21% for the baselines [2]. Yet the main benefit comes from learning to search using the EDFM [3], reaching to 59.7%.
> > > >
> > > > *Re “access to true world model”:* True, but the baselines had that too.  When evaluating Qi et al, and CEM, we gave them access to the true simulator, instead of a learned inaccurate forward model (Appendix D.4). Qi uses exhaustive search and fail since it can’t capture rich complex cascades, and because it is looking for a needle in a haystack.
> > > >
> > > > *Re “just an exhaustive search”:* That does not work. The number of possible cascades grows exponentially ( ~billions O(3^depth), Sec. 4.3) and it is infeasible to perform an exhaustive search. Searching efficiently in a tree, even with a perfect world model, is an open learning problem that triggered major AI successes such as AlphaGo and MuZero.
> > > >
> > > > *Reference to results:*
> > > >
> > > > [1] “EDFM alone” uses a naive all-or-none search. It is the Dirac-Delta result (33.5%) in Table 1 right.
> > > >
> > > > [2] Qi et al, and CEM results are presented in Table 1 left (21.1%, 20.9%).
> > > >
> > > > [3] “EDFM with learning to search” is the ROSETTE-MAX-L result (59.7%) in Table 1 right.

---

### Official Review · Reviewer_b7q6 · 2022-10-26

**Confidence:** 3
**Clarity, Quality, Novelty And Reproducibility:** 1. The dataset contains 46K scenes, h…
**Correctness:** 4
**Technical Novelty And Significance:** 3
**Empirical Novelty And Significance:** 3
**Recommendation:** 8

**Strength And Weaknesses:**

Strengths:
1. The paper is well written in terms of  introducing the new setup with good context.
2. The proposed learning setup, using event trees for the search space, and the scoring method – all seems very novel when combined together.

Weaknesses:
1. It would have been more interesting to see results on one more benchmark.


**Summary Of The Paper:**

The paper proposes a new learning setup called cascade in which an agent observes a dynamical system and then changes its initial state’s conditions to achieve a desired goal. The paper also proposes an interesting approach for learning a probabilistic scoring function over an “event tree” data structure to efficiently search for a complex cascade in a dynamic system. Bayesian formulation is used to leverage the observed cascade to guide the search. Experimental results show the usefulness of the proposed approach.

**Summary Of The Review:**

The paper presents an interesting and new learning setup with a novel learning approach. This seems like an exciting direction for the research community.

---

> ### Author Response · Authors · 2022-11-14
> **Response to reviewer b7q6**
>
> Thank you for the kind feedback and for your time!
>
> > **The dataset contains 46K scenes, how did you reach this number? Would adding more data helps the generalization?**
>
> We limited the compute time for generating scenes to 80 hours, and this yielded 46K scenes. It is likely that adding more data may improve generalization. An early experiment that we conducted with roughly half the data resulted in ~10% lower accuracy. We believe that generalization can also be further improved through better learning, e.g. by better negative examples. We updated the paper accordingly. See Section 5.1 (Dataset) and Appendix E (training data).

---

### Official Review · Reviewer_bYuU · 2022-11-03

**Confidence:** 2
**Correctness:** 3
**Technical Novelty And Significance:** 3
**Empirical Novelty And Significance:** 3
**Recommendation:** 6

**Clarity, Quality, Novelty And Reproducibility:**

The paper is clearly written, and novel in terms of both the problem setup and their method. The results should be largely reproducible.

**Strength And Weaknesses:**

Although I am not an expert on this problem of modeling cascading process, this paper is generally well-written and understandable to me. The following reviews are from the perspective of a person with general machine learning background.

Strengths:
1. This paper provides a precise formulation of the cascading process and the problem setup of making interventions in such systems.
2. They provide a testbed for this problem. Despite being simulated, it's a decent setup that people can follow and study the problem in a controlled and plausible way.
3. The method they proposed looks reasonable to me. They also compare with a bunch of baselines, some of which were proposed recently. So, the comparison looks convincing.
4. Regarding the experiments, they provide relatively comprehensive ablation studies and demonstration cases, which I think verifies their claims.

Weaknesses:
1. The scope of the cascading process can be very large, while the paper only focuses on a relatively simple simulation environment. It would be better to avoid overclaiming in the title or introduction.
2.  Although using the simulation environment is a reasonable setup right now, it's unclear how the studies in this toy setup can generalize to real-world scenarios. It might worth discussing at the end of the paper.

**Summary Of The Paper:**

This paper studies how to intervene in a dynamic system of cascading events, where events might trigger other events and lead to the “butterfly effect” in the end. Specifically, they propose a relatively simple simulation environment as a testbed for this problem - an agent observes a physical world with several moving and static objects and then manipulates the direction and speed of the purple ball to satisfy the complex set of constraints given by the instruction. They define the problem as a supervised learning setup, and then propose a method that uses tree search for modeling the forward process and employs machine learning to learn to search effectively in the exponential space. They present the superior performance of their method compared to several baseline models. They also provide various ablation studies and a human evaluation to demonstrate the effectiveness of their method.

**Summary Of The Review:**

The paper studies a new problem of intervening cascading process and proposes a supervised learning setup with a simulation environment as well as a better method for this problem. The paper looks convincing to me overall although I might not be very familiar with the literature of this work. I am leaning toward acceptance for this paper.

---

> ### Author Response · Authors · 2022-11-14
> **Response to reviewer bYuU**
>
> Thank you for the thoughtful feedback and for your time!
>
> > **The scope of the cascading process can be very large, while the paper only focuses on a relatively simple simulation environment. It would be better to avoid overclaiming in the title or introduction.**
>
> Thank you for helping us in refocusing the claims. Following this comment, we updated the title and introduction:
>
> * We renamed the title to “$\textnormal{Learning to Reason and Act in Cascading Processes }  \textcolor{purple}{\textnormal{Driven by Semantic Events}}$
> ”, to clarify the type of cascade processes we study.
>
> * We updated the introduction to stress that simulated test beds do not cover the full complexity of real-world scenarios.
>
> > **Although using the simulation environment is a reasonable setup right now, it's unclear how the studies in this toy setup can generalize to real-world scenarios. It might worth discussing at the end of the paper.**
>
> We agree that this is a very important point. Following this comment, we added a discussion of the “sim-to-real” path in this area” in Section 7. We believe that it can follow similar paths as in other areas of AI where approaches mature from toy datasets to realistic problems. In short, there are three main steps needed. First, creating a benchmark dataset for a real-life domain, annotated with semantic events. Some fields have datasets that can be very natural for the problem we discussed. These may include logistics (Appendix J), the evolution of natural disasters and their consequences (Zuccaro et al. 2018), and cascading failures in power grids (Schäfer et al. 2018)
>
> Second, using that dataset to train an event-driven forward model (EDFM). There are efficient approaches to learning such models from temporal data, like dynamic Bayes nets (Gharamani 1997, Bhattacharjya 2020, Gunawardana 2016), which can also handle latent variables.
> Finally, given the EDFM, our approach can be applied. Domain-specific properties can be used to improve the accuracy and robustness of an EDFM learned.
> With that said, we are encouraged that the reviewer stated “The simulation environment is a reasonable setup right now”. We feel that research in this area would currently benefit from using simulators and developing algorithms in controlled environments, before adding all complexities of real problems.
>
> > **The paper looks convincing to me overall although I might not be very familiar with the literature of this work.**
>
> As reviewer b7q6 wrote below, we also (biasedly :) ) think that this work is an “exciting direction for the research community.”. We hope that the rebuttal addresses all your questions.
>
>
> **References**
>
> Zuccaro et al. Theoretical model for cascading effects analyses, International Journal of Disaster Risk Reduction, 2018
>
> Schäfer et al. Dynamically induced cascading failures in power grids, Nature Communications, 2018
>
> Bhattacharjya et al. Event-driven continuous time bayesian networks, AAAI 2020
>
> Gunawardana et al. Universal Models of Multivariate Temporal Point Processes, AISTATS 2016
>
> Gharamani et al. Learning dynamic Bayesian networks, 1998

---

### Author Response · Authors · 2022-11-14
**General response to reviewers**

We thank the reviewers for their constructive feedback and for investing their time and effort. We uploaded a new version of the paper to address the reviewers' comments.

We are encouraged that the reviewers find the paper **novel** (all: $\textcolor{blue}{\textnormal{bYuU}}$, $\textcolor{green}{\textnormal{b7q6}}$, $\textcolor{magenta}{\textnormal{1vrT}}$), the setup and approach to be **very interesting** ($\textcolor{green}{\textnormal{b7q6}}$, $\textcolor{magenta}{\textnormal{1vrT}}$), and **“an exciting direction for the research community”** ($\textcolor{green}{\textnormal{b7q6}}$). They also found the experiments and claims to be **comprehensive** ($\textcolor{blue}{\textnormal{bYuU}}$), **well-supported and correct** ($\textcolor{blue}{\textnormal{bYuU}}$, $\textcolor{green}{\textnormal{b7q6}}$), with **convincing** comparisons to recent baselines ($\textcolor{blue}{\textnormal{bYuU}}$), and the paper to be **well-written** ($\textcolor{blue}{\textnormal{bYuU}}$, $\textcolor{green}{\textnormal{b7q6}}$) and **reproducible** ($\textcolor{blue}{\textnormal{bYuU}}$).


> **Some reviewers asked about future real-world applications of event trees and our approach**

Event trees and our approach may apply to a range of real-world scenarios that can be naturally modeled in an event-driven fashion. Appendix J originally discussed an application to Logistics. Following various comments by reviewers, we now added to the paper a discussion of two real-world scenarios in Section 7 and Appendix J, and discuss them briefly here.

First, Schäfer et al. (2018) studied cascading failures in power grids. Here, semantic events are failures of nodes (transformers, power generators, ...) or edges (power lines). The flow of power follows a known set of ODEs for a given grid (Eqs 14-15 in Schäfer). When flow exceeds the powerline capacity (Eqs 1-2), that line fails (an event), resulting in an effectively different grid and a different set of ODEs that govern the dynamics. The transmission system operator (TSO) may wish to define goals like “no more than three failures”, “no more than n people affected”, and “that highly important node must not fail”. An event-driven model is very natural for this scenario, and we expect our approach to provide a good solution. The intervention of the TSO may be intentional modifications to the topology of the grid such as “shut down a power line” or “increase auxiliary power in nodes X,Y, Z”.

Second, Zuccaro et al. (2018), developed event trees as a model of natural disasters and their consequences. They model transitions between events like a “seismic shock” which can lead to a “landslide” and result in a “traffic accident”, and taking preventive measures like “evacuate population”. Our approach can be used to efficiently make decisions in their system.


We address the additional concerns and remarks in individual replies to the reviewers

**References:**

Zuccaro et al. Theoretical model for cascading effects analyses, International Journal of Disaster Risk Reduction, 2018

Schäfer et al. Dynamically induced cascading failures in power grids, Nature Communications, 2018

---

### Decision · Program_Chairs · 2023-01-20

**Decision:**

Reject

**Justification For Why Not Higher Score:**

AC is not convinced that this is an important problem as formulated, while the significance of the algorithmic contributions remain unclear. One reviewer recommends 'accept' (8), however their review is scant on details, speaking only to a combination of search and scoring that *seems" novel.

**Justification For Why Not Lower Score:**

N/A

**Metareview: Summary, Strengths And Weaknesses:**

The paper considers the problem of controlling agents interacting with a dynamic environment in which events may have cascading effects. More specifically, the paper looks a setting in which an agent observes a system with known dynamics and a semantic instruction, and must intervene in order to trigger an appropriate cascade of events. The paper formulates this problem in the context of supervised learning, and proposes a method that uses semantic tree search to efficiently search exponentially large spaces. The method is shown to outperform several baseline methods in novel scenes, with ablations  and human evaluations that provide more insight into the framework.

The reviewers agree that the paper is well written and that the work is interesting with regards to the problem setup and the manner by which the proposed method learns a probabilistic scoring function over the event tree data structure to efficiently search complex cascades. However, there is notable disagreement among the reviewers in regards to the significance of the proposed framework in light of existing methods and the fact that the method is initially only evaluated on the proposed benchmark. The authors put considerable effort into addressing these concerns, both through further discussions to resolve the reviewers' questions/concerns, as well as the inclusion of additional results. This effort helps to clarify the significance of the paper's contributions, which as the authors state is the introduction of "a new problem for which state-of-the-art algorithms fail" together with a novel framework that takes advantage of the known structure of the problem to improve the efficiency of search. However, the AC does not see a clear recognition of the relevance of the problem itself, particularly in light of questions about how the study would generalize to the real world. While Reviewer b7q6 is positive (recommending 'Accept'), speaking to the perceived novelty of combining event trees with the scoring method, their review is otherwise noticeably short on details. Unfortunately, the AC was not able to get them to elaborate on their review. The AC also notes the authors' comments regarding Reviewer 1vrT and the reviewer's willingness to lower their confidence score to allow more weight to be placed on the other reviewers, though the lowered confidence now matches that of Reviewer b7q6.

**Summary Of Ac-Reviewer Meeting:**

N/A